# MAUVE: Measuring the Gap Between Neural Text and Human Text using Divergence Frontiers

**Krishna Pillutla**[1]    **Swabha Swayamdipta**[2]    **Rowan Zellers**[1]    **John Thickstun**[3]
**Sean Welleck**[1,2]    **Yejin Choi**[1,2]    **Zaid Harchaoui**[4]

[1]Paul G. Allen School of Computer Science & Engineering, University of Washington
[2]Allen Institute for Artificial Intelligence
[3]Department of Computer Science, Stanford University
[4]Department of Statistics, University of Washington

## Abstract

As major progress is made in open-ended text generation, measuring how close machine-generated text is to human language remains a critical open problem. We introduce MAUVE, a comparison measure for open-ended text generation, which directly compares the learnt distribution from a text generation model to the distribution of human-written text using divergence frontiers. MAUVE scales up to modern text generation models by computing information divergences in a quantized embedding space. Through an extensive empirical study on three open-ended generation tasks, we find that MAUVE identifies known properties of generated text, scales naturally with model size, and correlates with human judgments, with fewer restrictions than existing distributional evaluation metrics.

## 1 Introduction

Recent large-scale text generation models show an ability to produce human-like text of remarkable quality and coherence in open-ended generation [45, 61, 6]. In this setting, a text generation model forms a distribution over natural language sequences, induced by an autoregressive neural sequence model (e.g., GPT-3 [6]) paired with a decoding algorithm (e.g., nucleus sampling [26]). Generating text amounts to sampling from this distribution, with the goal of obtaining samples that resemble those from the "true" distribution of human-written text.

To evaluate how close a generation model's distribution is to that of human-written text, we must consider two types of errors: (I) where the model assigns high probability to sequences which do *not* resemble human-written text, and, (II) where the model distribution does not cover the human distribution, i.e., it fails to yield diverse samples. However, quantifying these aspects in a principled yet computationally tractable manner is challenging, as the text distributions are high-dimensional and discrete, accessed only through samples or expensive model evaluations [26, 58, 62].

We develop MAUVE, a comparison measure for open-ended text generation. The proposed measure is efficient, interpretable, and practical for evaluating modern text generation models. It captures both types of errors (Figure 1) by building upon *information divergence frontiers* [49, 31, 16], so far underexplored in natural language processing. The key idea for making the proposed measure computationally tractable, yet effective, is to reduce its measurement to computing Kullback-Leibler divergences in a quantized, low-dimensional space after embedding samples from each distribution with an external language model. From an end-user's perspective, MAUVE has a simple interface: given neural text and human text, it yields a scalar measure of the gap between them.

35th Conference on Neural Information Processing Systems (NeurIPS 2021).

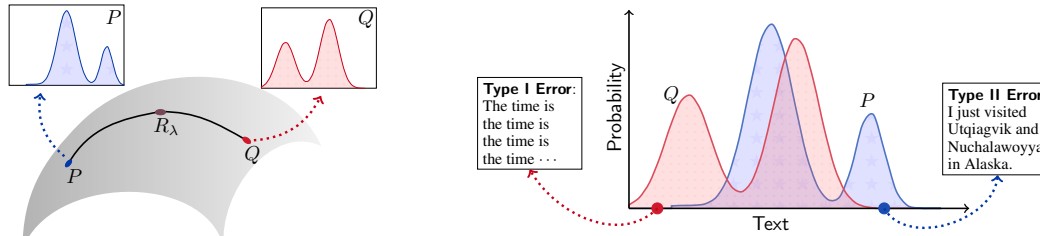

Figure 1: **Left**: MAUVE compares the machine text distribution $Q$ to that of human text $P$ by using the family of mixtures $R_\lambda = \lambda P + (1-\lambda)Q$ for $\lambda \in (0,1)$. **Right**: Illustration of *Type I errors*, where $Q$ produces degenerate, repetitive text which is unlikely under $P$, and, *Type II errors*, where $Q$ cannot produce plausible human text due to truncation heuristics [26]. MAUVE measures these errors softly, by using the mixture distribution $R_\lambda$. Varying $\lambda$ in $(0,1)$ gives a divergence curve and captures a spectrum of soft Type I and Type II errors. MAUVE summarizes the entire divergence curve in a single scalar as the area under this curve.

We summarize our contributions. First, we introduce MAUVE, a comparison measure between neural text and human text. Second, we empirically show that MAUVE is able to quantify known properties of generated text with respect to text length, model size, and decoding more correctly and with fewer restrictions than existing distributional evaluation metrics. Third, we find through a human evaluation that MAUVE better correlates with human quality judgements of text. Finally, we find that MAUVE can be highly robust to the choice of quantization, embeddings, and scaling. We open-source a pip-installable Python package to compute MAUVE.[1]

## 2  MAUVE

We begin by discussing the basics of open-ended text generation, and then introduce MAUVE for measuring the divergence between machine generated text and human text.

**Open-ended Text Generation.** A language model is an estimate $\hat{P}(\boldsymbol{x})$ of the probability distribution over sequences of text $\boldsymbol{x} = (x_1, \ldots, x_{|\boldsymbol{x}|})$, consisting of tokens $x_t$ belonging to a fixed vocabulary (e.g. characters, or words). Prevailing neural autoregressive language models estimate the joint distribution $\hat{P}(\boldsymbol{x})$ by modeling the conditional distribution $\hat{P}(x_{t+1}|\boldsymbol{x}_{1:t})$ over the next token in a sequence. The open-ended text generation task asks us to output text $\hat{\boldsymbol{x}}_{t+1:|\boldsymbol{x}|}$ in continuation of a given context $\boldsymbol{x}_{1:t}$. Unlike targeted generation tasks like translation or summarization, there is no "correct" output; the main criteria for open-ended text generation are coherence, creativity, and fluency.

Given a neural autoregressive language model $\hat{P}$, we can generate open-ended text in a serial, left-to-right fashion, by sampling $\hat{x}_{t+1} \sim \hat{P}(\cdot|\boldsymbol{x}_{1:t})$, $\hat{x}_{t+2} \sim \hat{P}(\cdot|\boldsymbol{x}_{1:t}, \hat{x}_{t+1})$, etc. In practice, this simple decoding algorithm is often modified by adjusting the conditional distribution $\hat{P}(\cdot|\boldsymbol{x}_{1:t})$ to promote more conservative outputs. The decoding algorithm and the language model taken together define a distribution $Q$ over text, which we call the *model distribution*. Common decoding algorithms include temperature rescaling [1] and truncation [18, 26]. Note that truncation methods in particular create sparsity in $Q$, which leads to degeneracy of some measures including test-set perplexity.

**Sources of Error in Text Generation.** Our goal in this work is to measure the gap between the model distribution $Q$ and the target distribution $P$ of human text. As highlighted in Figure 1, this gap arises from two sources of error:

(Type I)  $Q$ places high mass on text which is unlikely under $P$,
(Type II)  $Q$ cannot generate text which is plausible under $P$.

The Type I errors are false positives, including the common failure case where a model generates text with semantic repetitions [15, 26, 59] that are highly unlikely to be written by humans.[2] The Type II

---

[1] Available from `https://github.com/krishnap25/mauve`. See Appendix B for an example of the `mauve` package in action.

[2] Let text $\boldsymbol{x}$ with $P(\boldsymbol{x}) \gg 0$ be the positive class and $P(\boldsymbol{x}) \approx 0$ be the negative class. If $Q(\boldsymbol{x}) \gg 0$ for some negative $\boldsymbol{x}$, then the model incorrectly considers it a positive, so it is a *false* positive.

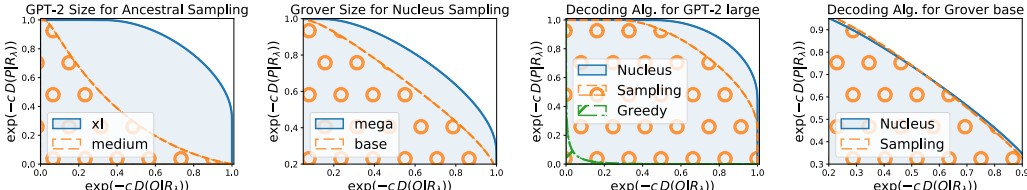

Figure 2: Divergence curves for different models (GPT-2 [45], Grover [61]) and decoding algorithms (greedy decoding, ancestral and nucleus sampling). MAUVE is computed as the area of the shaded region, and larger values of MAUVE indicate that $Q$ is closer to $P$. In general, MAUVE indicates that generations from larger models and nucleus sampling are closer to human text. **Rightmost**: Nucleus sampling has a slightly smaller Type I error than ancestral sampling but a higher Type II error, indicating that ancestral sampling with Grover base produces more degenerate text while nucleus sampling does not effectively cover the human text distribution.

errors are false negatives, which can occur, for instance, because some pieces of plausible human text cannot be generated by truncation-based decoding algorithms such as nucleus sampling [26]. The gap between $P$ and $Q$ is small only if both of these errors are small.

**Quantifying the Errors.** We formalize the Type I and II errors with the Kullback-Leibler (KL) divergences $\mathrm{KL}(Q|P)$ and $\mathrm{KL}(P|Q)$, respectively. The divergence $\mathrm{KL}(Q|P)$ penalizes $Q$ if there exists text $\boldsymbol{x}$ such that $Q(\boldsymbol{x})$ is large but $P(\boldsymbol{x})$ is small, so it quantifies the Type I error. Likewise, $\mathrm{KL}(P|Q)$ quantifies the Type II error.

Unfortunately, one or both of the KL divergences $\mathrm{KL}(P|Q)$ and $\mathrm{KL}(Q|P)$ are infinite if the supports of $P$ and $Q$ are not identical, which is often the case in open-ended generation. This makes the KL divergence itself unsuitable as an evaluation metric. We overcome this issue by *softly* measuring the two errors using the mixture distribution $R_\lambda = \lambda P + (1 - \lambda)Q$ for some $\lambda \in (0, 1)$. In particular, we define the (soft) Type I error at level $\lambda$ as $\mathrm{KL}(Q|R_\lambda)$ and the (soft) Type II error as $\mathrm{KL}(P|R_\lambda)$.

**Summarizing the Errors with a Divergence Curve.** Since the mixture weight $\lambda$ was arbitrary, we consider a family of Type I and II error values by varying $\lambda$ between 0 and 1, in the same spirit as information divergence frontiers [49, 16]. This yields a *divergence curve*,

$$\mathcal{C}(P, Q) = \left\{ \big( \exp(-c\,\mathrm{KL}(Q|R_\lambda)), \exp(-c\,\mathrm{KL}(P|R_\lambda)) \big) \,:\, R_\lambda = \lambda P + (1 - \lambda)Q, \lambda \in (0, 1) \right\}, \tag{1}$$

where $c > 0$ is a hyperparameter for scaling. The divergence curve formalizes and encodes information about the trade-off between Type I and II errors.[3] Figure 2 illustrates the divergence curves for different models and decoding algorithms.

Our proposed measure, **MAUVE**$(P, Q)$, is the area under the divergence curve $\mathcal{C}(P, Q)$. It provides a scalar summary of the trade-off between Type I and Type II errors. MAUVE$(P, Q)$ lies in $(0, 1]$, with a larger value meaning that $Q$ is closer to $P$. Further, MAUVE$(P, Q) = 1$ if and only if $Q = P$. The area under the curve is a common summary of trade-off curves in machine learning [13, 11, 12, 19].

**Connections to Common Divergences.** The divergence curve encodes more information than the KL divergence $\mathrm{KL}(P|Q)$, which can be obtained from the second coordinate of the curve $\mathcal{C}(P, Q)$ as $\lambda \to 0$, and the reverse KL divergence $\mathrm{KL}(Q|P)$ which can be obtained from the first coordinate of the curve $\mathcal{C}(P, Q)$ as $\lambda \to 1$. Further, the Jensen-Shannon (JS) divergence $\mathrm{JS}(P, Q) = \big(\mathrm{KL}(P|R_{1/2}) + \mathrm{KL}(Q|R_{1/2})\big)/2$, can be obtained from the two coordinates of $\mathcal{C}(P, Q)$ at $\lambda = 1/2$. MAUVE summarizes *all* of the divergence curve $\mathcal{C}(P, Q)$.

**Computing MAUVE for Open-Ended Text Generation.** Each point on the divergence curve $\mathcal{C}(P, Q)$ consists of a coordinate

$$\mathrm{KL}(P|R_\lambda) = \sum_{\boldsymbol{x}} P(\boldsymbol{x}) \log \frac{P(\boldsymbol{x})}{R_\lambda(\boldsymbol{x})}, \tag{2}$$

and a similarly defined coordinate $\mathrm{KL}(Q|R_\lambda)$. We cannot compute the summation as written in Eq. (2), as we do not know the ground-truth probabilities $P(\cdot)$ and the support of a typical model

---

[3]More generally, the divergence curve $\mathcal{C}(P, Q)$ encodes the **Pareto frontier** of $\big(\mathrm{KL}(P|R), \mathrm{KL}(Q|R)\big)$ for all distributions $R$, not just mixtures of the form $R_\lambda$. We prove this in Appendix A.

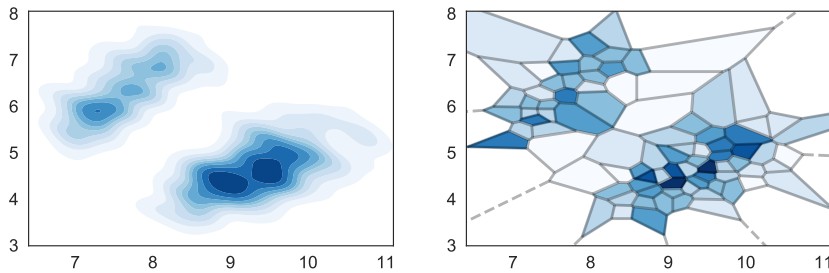

Figure 3: Illustration of the quantization. **Left**: A continuous two-dimensional distribution $P$. **Right**: A partitioning of the Euclidean plane $\mathbb{R}^2$ and the corresponding quantized distribution $\tilde{P}$.

distribution is prohibitively large, since it is the space of all sequences of tokens. As a result of these two issues, MAUVE cannot be tractably computed in closed form.

We employ a Monte Carlo estimator using samples $\boldsymbol{x}_i \sim P$ and $\boldsymbol{x}_i' \sim Q$ to overcome the fact that ground-truth probabilities $P(\cdot)$ are unknown. We circumvent the intractable support size by computing MAUVE in a quantized embedding space that is sensitive to important features of text.

The overall estimation procedure is as follows. First, we sample human text $\boldsymbol{x}_i \sim P$ and machine text $\boldsymbol{x}_i' \sim Q$. We then embed each text sequence using an external language model $M$ (e.g., GPT-2 [45]) to obtain embeddings $\{M(\boldsymbol{x}_i)\}_{i=1}^N$ and $\{M(\boldsymbol{x}_i')\}_{i=1}^{N'}$. Each embedding is now a vector $M(\boldsymbol{x}) \in \mathbb{R}^d$. Next, we jointly quantize the embedded samples (e.g. with $k$-means [36]), and count the cluster assignments to form histograms, giving low-dimensional discrete distributions that approximate each high-dimensional text distribution. In particular, the distribution $P$ of human text is approximated by the discrete distribution $\tilde{P}$ of support size $k$, which is defined as,

$$\tilde{P}(j) = \frac{1}{N} \sum_{i=1}^{N} \mathbb{I}\big(\phi(\boldsymbol{x}_i) = j\big),\tag{3}$$

where $\phi(\boldsymbol{x}) \in \{1, \cdots, k\}$ returns the cluster id of $\boldsymbol{x}$. The model distribution $Q$ is approximated as $\tilde{Q}$ similarly. Here, $\tilde{P}$ and $\tilde{Q}$ can be interpreted as piecewise constant approximations of $P$ and $Q$, similar to a histogram; see Figure 3 for an illustration. Computing the divergence curve is now tractable, as each coordinate is a KL divergence between the $k$-element discrete distributions.

To recap, our proposed measure **MAUVE**$(P, Q)$ is the area under this divergence curve, providing a summary of all Type I and Type II errors through an efficient approximation designed for text generation. Next, we discuss how MAUVE compares to prior comparison measures for text (§3), then present empirical results with MAUVE (§4).

## 3  Related Work

**Divergence Measures for Text.** Prior measures of similarity/divergence between machine text and human text come in three broad categories: (a) reference-based, (b) statistics-based, and (c) language modeling. Table 1 summarizes the latter two categories, and contrasts them with MAUVE.

*Reference-based measures* evaluate generated text with respect to a (small set of) reference text sample(s), rather than comparing full sequence distributions. These include classical metrics for $n$-gram matching [44, 32, 2], which are designed to capture similarities in the surface form of the generated text and the human references, making them fundamentally ill-suited for open-ended generation. Moreover, it has been recently shown in [42] show that these classical metrics only weakly agree with human judgments.

More recent reference-based metrics are capable of comparisons in a high dimensional space [53, 63, 51, 9], thereby capturing distributional semantics beyond superficial $n$-gram statistics. For instance, Moverscore [64] relies on the Word Mover's distance [30], and is an instance of an optimal transportation distance [57]. It computes the minimum cost of transforming the generated text to the reference text, taking into account Euclidean distance between vector representations of $n$-grams,

| Type | Metric | Measures | Approximates |
|------|--------|----------|--------------|
| Statistics | Zipf Coefficient [26]
Self-BLEU [65]
Generation Perplexity [18] | Unigram rank-frequency statistics
N-gram diversity
Generation quality via external model $R$ | –
–
$\lvert \mathbb{E}_Q[\log R(\boldsymbol{x})] - \mathbb{E}_P[\log R(\boldsymbol{x})]\rvert$
(a single point inside $\mathcal{C}(P,Q)$) |
| Language Modeling | Perplexity
$\varepsilon$-perplexity [39]
Sparsemax Score [39]
Token JS-Div. [39] | Test-set perplexity
Perplexity w/ Laplace smoothing
LM quality (sparsemax loss [38])
LM quality (JS divergence) | $\mathbb{E}_P[\log Q(\boldsymbol{x})]$
$\mathbb{E}_P[\tilde{Q}(\boldsymbol{x})]$
$\mathbb{E}_P[\tilde{Q}(\boldsymbol{x})]$
$\mathbb{E}_P[\tilde{Q}(\boldsymbol{x})]$ |
| Divergence Curve | MAUVE (this work) | Quality & diversity via the divergence curve | $\mathcal{C}(P,Q)$ at all $\lambda$ |

Table 1: Summary of automatic distributional metrics for evaluating open-ended text generation. MAUVE provides a summary of all points along the divergence curve, rather than a single point. The summary is based on comparisons in a joint embedding space, rather than a statistic computed independently on each distribution. $\tilde{Q}$ informally refers to a quantity related to $Q$.

as well as their document frequencies. The paradigm of reference-based measures is useful for targeted generation tasks such as translation and summarization where matching a set of references is paramount. It is, however, unsuitable for open-ended generation where there typically are several plausible continuations for each context and creative generations are desirable.

*Statistics-based measures* compare the model distribution $Q$ with respect to the human distribution $P$ on the basis of some statistic $T(P)$ and $T(Q)$. Property-specific statistics such as the amount of repetition [26, 59], verifiability [40], or termination [58] are orthogonal to MAUVE, which provides a summary of the overall gap between $P$ and $Q$ rather than focusing on an individual property. Another statistic is the generation perplexity [18, 26], which compares the perplexity of the model text $\boldsymbol{x} \sim Q$ with that of human text $\boldsymbol{x}' \sim P$ under an external model $R$. By virtue of $T(\cdot)$ being a scalar, generation perplexity cannot trade-off the Type I and Type II errors like MAUVE. In fact, we show in Appendix A that the generation perplexity can be derived from *a single point* enclosed between the divergence curve and the axes.

*Language modeling metrics* calculate how (un)likely human text $\boldsymbol{x} \sim P$ is under the model distribution $Q$, for instance, using the probability $Q(\boldsymbol{x})$. These metrics are related to a single point on the divergence curve, rather than a full summary. Examples include the perplexity of the test set (which is a sample from $P$) under the model $Q$ and its generalizations to handle sparse distributions [39]. Unlike MAUVE, these metrics never see model text samples $\boldsymbol{x}' \sim Q$, so they cannot account for how likely the model text is under the human distribution $P$. Moreover, they cannot be used for decoding algorithms such as beam search which do not define a token-level distribution.

Automatic metrics have been proposed for specific domains such as generation of dialogues [55], stories [21], and others [43]. They capture task-specific properties; see the surveys [8, 48]. In contrast, MAUVE compares machine and human text in a domain-agnostic manner. Other related work has proposed metrics that rely on multiple samples for quality-diversity evaluation [7], and Bayesian approaches to compare the distribution of statistics in machine translation [17].

**Non-automatic Metrics.** HUSE [24] aims to combine human judgements of Type I errors with Type II errors measured using perplexity under $Q$. Due to the costs of human evaluation, we consider HUSE, as well other metrics requiring human evaluation, such as single-pair evaluation, as complementary to MAUVE, which is an automatic comparison measure. As a separate technical caveat, it is unclear how to use HUSE for sparse $Q$ that assigns zero probability to a subset of text, which is the case with state-of-the-art decoding algorithms [26, 39].

**Evaluation of Generative Models.** Evaluation of generative models is an active area of research in computer vision, where generative adversarial networks [20] are commonly used. However, metrics such as Inception Score [50] are based on large-scale supervised classification tasks, and thus inappropriate for text generation. The Fréchet Distance [25, 52] and its unbiased counterpart, the Kernel Inception Distance [5] are both used for evaluating generative models, but unlike MAUVE, do not take into account a trade-off between different kinds of errors between the learned and a

| Task Domain | Model | Finetuning | Dataset | Prompt Length | Max. Generation Length | Number of Generations |
|---|---|---|---|---|---|---|
| Web text | GPT-2 (all sizes) | Pretrained | Webtext | 35 tokens | 1024 tokens | 5000 |
| News | Grover (all sizes) | Pretrained | RealNews | varying | 1024 tokens | 5000 |
| Stories | GPT-2 medium | Finetuned | WritingPrompts | 50 tokens | 512 tokens | 5000 |

Table 2: Dataset and task summary. Note that 1024 tokens correspond to $\sim 750$ words on average.

reference distribution. Sajjadi et al. [49] and Kynkäänniemi et al. [31] both proposed metrics based on precision-recall curves. Djolonga et al. [16] proposed information divergence frontiers as a unified framework emcompassing both these works as special cases. MAUVE extends the above line of work, and is operationalized for open-ended text generation, applicable for data generated by large-scale neural language models. Complementary to this work, Liu et al. [33] study the theory of information divergence frontiers, proving non-asymptotic bounds on the estimation and quantization error.

## 4 Experiments

We perform three sets of experiments to validate MAUVE. Our first set of experiments (§4.1) examine how known properties of generated text with respect to generation length, decoding algorithm, and model size can be identified and quantified by MAUVE. Next, in §4.2 we demonstrate that MAUVE is robust to various embedding strategies, quantization algorithms, and hyperparameter settings. Finally, in §4.3 we find that MAUVE correlates with human judgments. The code as well as the scripts to reproduce the experiments are available online.[4]

**Tasks.** We consider open-ended text generation using a text completion task [26, 59] in three domains: web text, news and stories. Each domain consists of a sequence dataset split into (context, continuation) pairs. Given a context $\boldsymbol{x}_{1:k}$, the task is to generate a continuation $\hat{\boldsymbol{x}}_{k+1:T} \sim Q(\cdot \mid \boldsymbol{x}_{1:k})$, forming a completion. Each ground-truth completion $\boldsymbol{x}_{1:T}$ is considered a sample from the true distribution $P$, while the completion $(\boldsymbol{x}_{1:k}, \hat{\boldsymbol{x}}_{k+1:T})$ is considered a sample from $Q$. The datasets, context and completion lengths, and number of completions used for each domain are shown in Table 2.

**Models.** As the language model $\hat{P}(\cdot)$, we use GPT-2, a large-scale transformer [56] pretrained on the web text dataset (see [45]), that is representative of state-of-the-art autoregressive language models. As the embedding model $M(\cdot)$ we use GPT-2 Large, and compare others in §4.2.

**Decoding Algorithms.** We consider three common decoding algorithms: *ancestral sampling* which samples directly from the language model's per-step distributions, $x_t \sim \hat{P}(x_t \mid \boldsymbol{x}_{1:t})$, *greedy decoding* which selects the most likely next token, $x_t = \arg\max_{x \in \mathcal{V}} \hat{P}(x \mid \boldsymbol{x}_{1:t})$, as well as *nucleus sampling* [26] which samples from top-$p$ truncated per-step distributions, $x_t \sim \hat{P}_{\text{nuc},p}(x_t \mid \boldsymbol{x}_{1:t})$, which is defined as

$$\hat{P}_{\text{nuc},p}(x_t \mid \boldsymbol{x}_{1:t}) \propto \begin{cases} \hat{P}_{\text{nuc},p}(x_t \mid \boldsymbol{x}_{1:t}), & \text{if } x_t \in V_p, \\ 0, & \text{else.} \end{cases}$$

Here, the top-$p$ vocabulary $V_p$ is the smallest set $V$ such that $\sum_{x \in V} \hat{P}(x \mid \boldsymbol{x}_{1:t}) \geq p$.

We also consider an adversarial sampling procedure, designed to generate low-quality text that nevertheless matches the perplexity of human text. Adversarial perplexity sampling proceeds in two phases: (1) we generate the first 15% of tokens in a sequence uniformly at random from the vocabulary, and (2) we generate the remaining tokens greedily to make the running perplexity of the generated sequence as close as possible to the perplexity of human text.

### 4.1 Quantifying Properties of Generated Text

To study MAUVE's effectiveness as a measure for comparing text distributions, we first examine how MAUVE quantifies known properties of generated text: a good measure should meet expected behavior that is known from existing research on each property. Specifically, we investigate how MAUVE behaves under changes in generation length, decoding algorithm, and model size.

---

[4]https://github.com/krishnap25/mauve-experiments.

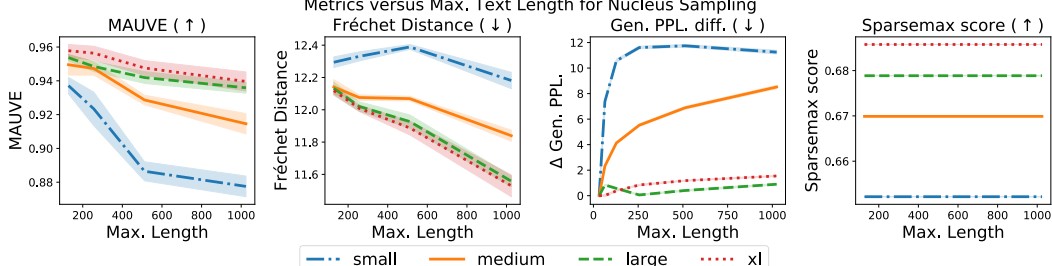

Figure 4: Generation quality versus maximum generation length according to MAUVE and three alternative measures (web text, GPT-2). MAUVE is the only comparison measure which identifies that generation quality decreases monotonically with increasing text length. The shaded area shows one standard deviation over generations from 5 random seeds.

**MAUVE quantifies quality differences due to generation length.** Although large transformer-based models can generate remarkably fluent text, it has been observed that the quality of generation deteriorates with text length: as the generation gets longer, the model starts to wander, switching to unrelated topics and becoming incoherent [46]. As a result, an effective measure should indicate lower quality (e.g. lower MAUVE) as generation length increases.

Figure 4 shows MAUVE as the generation length increases, along with three alternative metrics: generation perplexity, sparsemax score, and Fréchet distance [25, 52]. MAUVE reflects the desired behavior, showing a decrease in quality (lower MAUVE) as generation length grows, with the trend consistent across model sizes. The other three metrics, however, show less favorable trends. Fréchet distance indicates *improving* quality as the length increases, while generation perplexity shows non-monotonic quality trends for the small and large models. Finally, language modeling metrics such as the sparsemax score [39] remain constant, since they do not depend on the samples generated.

**MAUVE identifies quality differences between decoding algorithms.** Recent work has identified two clear trends in open-ended text generation with standard autoregressive models: (1) using greedy decoding results in repetitive, degenerate text [26, 59, 58]; (2) nucleus sampling (and related truncated sampling methods) yields higher quality text than ancestral sampling [18, 26].[5] An effective measure should thus indicate the quality relationship greedy $\prec$ ancestral $\prec$ nucleus.

Table 3 summarizes MAUVE's quality measures of greedy decoding, ancestral sampling, and nucleus sampling, along with alternative automated metrics and a human quality score. MAUVE correctly identifies the expected quality relationship, assigning the lowest quality to greedy decoding (.016) followed by ancestral sampling (.882), and the highest quality to nucleus sampling (.940). Other commonly-used metrics fail to identify this relationship: generation perplexity rates the highly degenerate greedy-decoded text as better than ancestral sampling (11.324 vs. 19.284), while the language-modeling metrics (SP, JS, $\varepsilon$-PPL) rate nucleus-decoded text as equal to or worse than greedy decoding or ancestral sampling. Further, as we show in Appendix D, MAUVE rightly identifies degeneracy of beam search, thus quantifying the qualitative observations of Holtzman et al. [26]. Finally, generation perplexity falls victim to the adversarial decoder (Adv.), unlike MAUVE.[6]

**MAUVE quantifies quality differences due to model size.** Scaling the model size has been a key driver of recent advances in NLP, with larger models leading to better language modeling and higher quality generations in open-ended settings [45, 6]. An effective metric should capture the relationship between model size and generation quality, which we verify with human quality scores.

Table 4 shows MAUVE's quality measures as the model size increases, along with alternatives and human quality scores. MAUVE increases as model size increases, agreeing with the human quality measure and the expectation that larger models should have higher quality generations. The widely-used generation perplexity, however, incorrectly rates the large model's text as the best. Although the language modeling metrics (SP, JS, and $\varepsilon$-PPL) capture the size-quality relationship, they are constant with respect to length (Figure 4), and did not correctly quantify decoding algorithm quality (Table 3).

---

[5]In general this relationship depends on the nucleus hyperparameter $p$ and task. Here, we follow the same settings as Holtzman et al. [26], and additionally include a human-assessed measure of quality.

[6]The results are consistent across model sizes and random seeds (see Appendix D).

| | Adv. | Greedy | Sampling | Nucleus |
|---|---|---|---|---|
| **Gen. PPL**($\downarrow$) | **0.05** | 11.3 | 19.3 | 1.54 |
| **Zipf**($\downarrow$) | 0.03 | 0.02 | 0.02 | **0.01** |
| **Self-BLEU**($\downarrow$) | 0.07 | 0.03 | **0.02** | 0.03 |
| **SP**($\uparrow$) | – | 0.50 | **0.69** | 0.69 |
| **JS**($\downarrow$) | – | **0.35** | 0.37 | 0.36 |
| $\varepsilon$-**PPL**($\downarrow$) | – | 497 | **11.4** | 13.7 |
| **MAUVE** ($\uparrow$) | 0.06 | 0.02 | 0.88 | **0.94** |
| **Human**($\uparrow$) | – | – | 9.0 | **15.7** |

| | Small | Medium | Large | XL |
|---|---|---|---|---|
| **Gen. PPL**($\downarrow$) | 11.2 | 8.5 | **0.9** | 1.5 |
| **Zipf**($\downarrow$) | 0.06 | **0.00** | 0.02 | 0.01 |
| **Self-BLEU**($\downarrow$) | 0.05 | **0.02** | 0.03 | 0.03 |
| **SP**($\uparrow$) | 0.65 | 0.67 | 0.68 | **0.69** |
| **JS**($\downarrow$) | 0.41 | 0.39 | 0.37 | **0.36** |
| $\varepsilon$-**PPL**($\downarrow$) | 25.9 | 18.8 | 14.9 | **13.7** |
| **MAUVE** ($\uparrow$) | 0.878 | 0.915 | 0.936 | **0.940** |
| **Human**($\uparrow$) | −15.9 | −3.4 | 12.6 | **15.7** |

Table 3: Generation quality w.r.t different **decoding algorithms** (web text, GPT-2 xl) under various metrics, and humans. MAUVE correctly captures the relationship greedy $\prec$ ancestral $\prec$ nucleus, and rates the adversarial decoder's text as low quality. Results are consistent across model sizes and random seeds. Boldfaced/highlighted entries denote the best decoding algorithm under each metric.

Table 4: Generation quality w.r.t different **model sizes** (web text, nucleus sampling) under various metrics, as well as human evaluators. MAUVE captures the relationship between model size and generation quality, agreeing with human-evaluated quality. Results are consistent across random seeds and decoding algorithms. Boldfaced/highlighted entries denote the best model size under each metric.

Table 6 in Appendix D shows additional results with ancestral sampling. In this case, human evaluators rated generations from the small model as better than those from the medium model. Interestingly, MAUVE also identified this relationship, agreeing with the human ratings, in contrast to the other automatic metrics we surveyed.

**Summary.** MAUVE identifies properties of generated text that a good measure should capture, related to length, decoding algorithm, and model size. In contrast, commonly used language modeling and statistical measures did not capture all of these properties. Unlike these alternatives, which capture a single statistic or relate to a single point on the divergence curve, MAUVE's summary measure incorporates type I errors that quantify the degenerate text produced by greedy decoding (recall Figure 1), while capturing distribution-level information that describes quality changes from generation length, model size, and the nuanced distinction between ancestral and nucleus sampling.

### 4.2 Approximations in MAUVE

MAUVE summarizes the divergence between two text distributions with an approximation that relies on two components: an embedding model $M(\boldsymbol{x})$ and a quantization algorithm $\mathcal{A}$ (§2, Eq. (3)). We study the effects of these two components.

**MAUVE works with alternative embedding models.** Figure 5 (left) shows that MAUVE with features from RoBERTa- large [34] gives qualitatively similar trends across model size and decoding as MAUVE with features from GPT-2 large. Quantitatively, the Spearman rank correlation between them across all model and decoders is 0.993. We observe that RoBERTa penalizes smaller models more than GPT-2 but rates greedy decoding higher. We leave further study of inductive biases in the different embedding models to future work.

**MAUVE is robust to quantization.** We compare different three different quantization algorithms:

(a) $k$-Means: We cluster the hidden representations using $k$-means, and represent them by their cluster membership to get a discrete distribution with size equal to the number of clusters.

(b) Deep Residual Mixture Models (DRMM): As a generalization of $k$-means, we train a deep generative model known as DRMM [22]. We convert the soft clustering returned by DRMM into a hard clustering by assigning each point to its most likely cluster, and quantize the data using the cluster membership. We use DRMM with 3 layers and 10 components per layer for a total of $10^3$ clusters, and train it for 20 epochs.

(c) Lattice Quantization: We learn a 4-dimensional feature representation of the vectors $M(\boldsymbol{x})$ using a deep network which maintains the neighborhood structure of the data while encouraging the features to be uniformly distributed on the unit sphere [47]. We quantize the data on a uniform lattice into 744 bins.

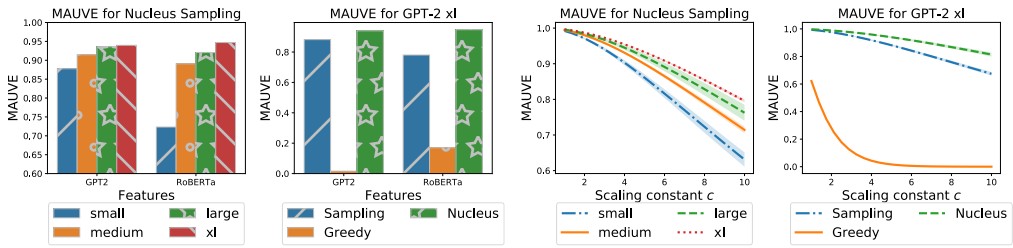

Figure 5: **Left**: MAUVE computed using GPT-2 (default) and RoBERTa [34] embeddings, across model sizes and decoding algorithms; see Table 12 in the Appendix for further results. The Spearman rank correlation between the two is **0.993** across model sizes and decoding algorithms. **Right**: Effect of the scaling constant $c$ on MAUVE. Choice of $c$ does not affect the relative order of the curves but only the numerical value. We use $c = 5$ to get interpretable values with both nucleus and greedy decoding.

We compare different choices of the quantization to $k$-means with $k = 500$, which is our default. The Spearman rank correlation between MAUVE computed with $k$-means for $k$ ranging from 100 to 5000 correlates nearly perfectly with that of $k = 500$. In particular, the Spearman correlation is exactly 0.99 or 1.00. Likewise, MAUVE computed with DRMM or lattice quantization has a near-perfect Spearman correlation of at least 0.99 with $k$-means. While the actual numerical value of MAUVE could vary with the quantization algorithm, these results show that the *rankings induced by various variants of* MAUVE *are nearly identical*.

**Practical recommendation for scaling parameter.** Figure 5 (right) shows the effects of adjusting the scaling parameter $c$, which does not affect the relative order of the divergence curve, but adjusts the numerical value returned by MAUVE. As a practical recommendation, we found $c = 5$ to yield interpretable values.

### 4.3 Correlation with Human Judgments

An effective metric should yield judgments that correlate highly with human judgments, assuming that human evaluators represent a gold-standard.[7] We evaluate how MAUVE's quality judgments correlate with human quality judgments. In our study, a quality judgment means choosing a particular (model, decoder) setting based on the resultant generations.

**Evaluation Protocol.** To obtain human judgments, we employ a pairwise setup: at each round, an annotator receives a context and continuations from two different (model, decoder) settings, and selects the continuation they found more natural using a 5-point Likert scale. Our interface for collecting annotations is shown in Figure 9 of Appendix E, which also includes further details and additional results.

We collect these annotations for web text generation with 8 different (model, decoder) settings plus a ninth setting for human-written continuations. Each setting is a GPT-2 model size paired with either ancestral or nucleus sampling. This gives us a total of 36 pairs of settings. Given the known difficulties with human evaluation of longer texts [28], we use a maximum completion length of 256 tokens. We obtain 90 preference ratings for each pair of settings, coming from a total of 214 crowd-workers from the Amazon Mechanical Turk platform. The evaluators were paid USD 0.40 per evaluation based on an estimated wage of USD 16 per hour.

We convert these pairwise preferences to a ranking by fitting a Bradley-Terry model [37], a parametric model used to predict the outcome of a head-to-head comparison. In particular, we obtain a score $w_i$ for each setting $i$ so that the log odds of humans preferring setting $i$ to setting $j$ in a head-to-head comparison is given by the difference $w_i - w_j$. For a given comparison measure, we compute the Spearman rank correlation between the comparison measure and the fitted Bradley-Terry coefficients $w_i$ for each of the (model, decoder) settings. The end result is a correlation score in $[-1, 1]$, with higher values meaning that quality judgments using the comparison measure correlate with quality judgments made by human evaluators.

---

[7]Concurrent work has shown that human evaluation might not always be consistent [10, 29]; however human judgments continue to be the gold standard for evaluating open-ended text generation.

| Metric | Task | Gen. PPL | Zipf Coef. | REP | Distinct-4 | Self-BLEU | Mauve |
|--------|------|----------|-----------|-----|-----------|-----------|-------|
| Human-like/BT | Web text | 0.810 | 0.833 | −0.167 | 0.738 | 0.595 | **0.952** |
| Interesting/BT | Web text | 0.643 | 0.524 | −0.143 | 0.524 | 0.405 | **0.810** |
| Sensible/BT | Web text | 0.738 | 0.690 | −0.071 | 0.595 | 0.524 | **0.857** |
| % Disc. Acc. | News | 0.468 | 0.595 | 0.792 | 0.653 | 0.516 | **0.956** |
| % Disc. Acc. | Stories | 0.643 | 0.643 | 0.250 | 0.750 | 0.857 | **0.893** |

Table 5: Correlation of various similarity measures with human judgments when available, and the accuracy of a trained discriminator otherwise. "BT" denotes the Bradley-Terry score for a pairwise human evaluation (§ 4.3). Boldfaced/highlighted numbers indicate highest correlation in each row. We observe that MAUVE has the highest correlation with human evaluation and discriminator accuracy.

**MAUVE correlates with human judgments.** Table 5 shows the correlation between human judgments and five automatic evaluation metrics obtained using our evaluation protocol on the web text domain. MAUVE correlates highly with human judgments of how human-like (0.952), interesting (0.810), and sensible (0.857) the machine text is. MAUVE's correlations with human judgments are substantially higher than those for the other automated measures; for instance, the commonly-used generation perplexity has correlations that are 0.12 to 0.17 lower than MAUVE's. The results suggest that MAUVE may act as an effective, automatic surrogate for costly human judgments.

**MAUVE correlates with learned discriminators.** We also measure the quality of generations by how well a trained model (a discriminator) can distinguish between real and generated text [35]. We report the test accuracy of a binary classifier trained to discriminate between machine and human text; a lower discrimination accuracy implies that the generation is harder to distinguish from human text. We report the accuracy of Grover mega as the discriminator for the news generations as it produced the highest discrimination accuracy [61] while we use GPT-2 large for the story domain. As seen in Table 5, MAUVE correlates the highest with the discrimination accuracy (0.96 for news and 0.89 for stories) among all comparison measures. Computing the discrimination accuracy for each (model, decoder) pair requires fine-tuning a separate model, which is particularly expensive for large models such as Grover-mega. MAUVE, on the other hand, does not require any training.

## 5    Conclusion

We presented MAUVE, an automatic measure of the gap between neural text and human text for open-ended text generation. MAUVE measures the area under a divergence curve, formalizing and summarizing a spectrum of errors that capture phenomena present in machine and human-generated text. MAUVE correlated with human judgments and identified quality differences due to generation length, decoding algorithm, and model size, which prior metrics struggle to capture. Automated metrics have driven advances in computer vision and many other machine learning domains. MAUVE's principled foundation and strong empirical performance offers a similar path forward for open-ended text generation systems. Extensions of MAUVE to closed-ended tasks, such as summarization and translation, where generations must be compared to a fixed set of gold-standard references, are promising directions for future work.

**Broader Impacts Statement**    MAUVE rewards model text which resembles human-authored text. However, we acknowledge the risks of rewarding systems that try to mimic humans [4], which is the ultimate goal of open-ended text generation. While our research is important for developing better language generators, we also encourage the community to pay attention to the development of technology that can reliably distinguish between human and machine text. We leave the extension of our method towards building such systems to future work.

**Acknowledgments**    Part of this work was done while Zaid Harchaoui was visiting the Simons Institute for the Theory of Computing, and while John Thickstun was at the University of Washington. This work was supported by NSF DMS-2134012, NSF CCF-2019844, NSF DMS-2023166, the DARPA MCS program through NIWC Pacific (N66001-19-2-4031), the CIFAR "Learning in Machines & Brains" program, a Qualcomm Innovation Fellowship, and faculty research awards.

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
