# Appendix

## Table of Contents

# A Divergence Curves and Mauve: Additional Details

We discuss some aspects of the divergence curves alluded to in §2 and §3. In particular, we discuss the following.

- Appendix A.1: the Pareto optimality of the divergence curves, mentioned in a footnote in §2.
- Appendix A.2: the connection between generation perplexity and the divergence curves as mentioned in §3.
- Appendix A.3: a formal definition of the quantization which is first introduced in §2, as well as an illustration.
- Appendix A.4: the pesudocode for MAUVE.

## A.1 Pareto Optimality of Divergence Curves

Here, we show the property of Pareto optimality of $\mathcal{C}(P, Q)$. We refer to the textbook [23] for more background on information theory and KL divergence. The main property we will show in this section is the following.

**Proposition 1.** *Consider two distributions $P, Q$ with finite support and a scaling constant $c > 0$. Let $R_\lambda$ be such that $\left(e^{-c\,\mathrm{KL}(Q|R_\lambda)}, e^{-c\,\mathrm{KL}(P|R_\lambda)}\right) \in \mathcal{C}(P, Q)$. Then, $R_\lambda$ is Pareto-optimal for the pair of objectives $\left(\mathrm{KL}(Q|\cdot), \mathrm{KL}(P|\cdot)\right)$. In other words, there does* not *exist any distribution $R$ such that $\mathrm{KL}(Q|R) < \mathrm{KL}(Q|R_\lambda)$ and $\mathrm{KL}(P|R) < \mathrm{KL}(P|R_\lambda)$ simultaneously.*

*Proof.* Let $\mathcal{F}(P, Q)$ be the Pareto frontier of $\left(\mathrm{KL}(Q|\cdot), \mathrm{KL}(P|\cdot)\right)$. The convexity of $\mathrm{KL}(Q|\cdot), \mathrm{KL}(P|\cdot)$ allows us to compute the Pareto frontier $\mathcal{F}(P, Q)$ exactly by minimizing linear combinations of the objectives. Concretely, we have from [41, Thm. 3.4.5, 3.5.4] that

$$\mathcal{F}(P, Q) = \left\{ \left(\mathrm{KL}(P|R_\lambda^\star), \mathrm{KL}(P|R_\lambda^\star)\right) \; : \; \lambda \in [0, 1] \right\}$$

where

$$R_\lambda^\star \in \arg\min_R \{\lambda\,\mathrm{KL}(Q|R) + (1 - \lambda)\mathrm{KL}(P|R)\}\,.$$

We invoke the next lemma to show that $R_\lambda^\star = \lambda P + (1 - \lambda)Q$ to complete the proof. $\qquad\square$

**Lemma 2.** *Let $P, Q, S$ be discrete distributions with finite support. For any $\lambda \in [0, 1]$ and $\bar{\lambda} = 1 - \lambda$, letting $R_\lambda = \lambda P + \bar{\lambda} Q$, we have the identity*

$$\lambda\,\mathrm{KL}(P|S) + \bar{\lambda}\,\mathrm{KL}(Q|S) = \lambda\,\mathrm{KL}(P|R_\lambda) + \bar{\lambda}\,\mathrm{KL}(Q|R_\lambda) + \mathrm{KL}(R_\lambda|S)\,.$$

*Consequently, we have that*

$$R_\lambda \in \arg\min_S \left\{\lambda\,\mathrm{KL}(P|S) + \bar{\lambda}\,\mathrm{KL}(Q|S)\right\}\,.$$

*Proof.* By adding and subtracting $\sum_i R_{\lambda,i} \log(R_{\lambda,i})$, we get,

$$\lambda\,\mathrm{KL}(P|S) + \bar{\lambda}\,\mathrm{KL}(Q|S) = \sum_i \lambda P_i \log P_i + \bar{\lambda} Q_i \log Q_i - R_{\lambda,i} \log S_i$$

$$= \sum_i \lambda P_i \log \frac{P_i}{R_{\lambda,i}} + \bar{\lambda} Q_i \log \frac{Q_i}{R_{\lambda,i}} + R_{\lambda,i} \log \frac{R_{\lambda,i}}{S_i}$$

$$= \lambda\,\mathrm{KL}(P|R_\lambda) + \bar{\lambda}\,\mathrm{KL}(Q|R_\lambda) + \mathrm{KL}(R_\lambda|S)\,.$$

The first two terms are independent of $S$ and the last term is minimized at $S = R_\lambda$. $\qquad\square$

**Connection to Divergence Frontiers [16].** The Pareto frontier $\mathcal{F}(P, Q)$ of $\left(\mathrm{KL}(Q|\cdot), \mathrm{KL}(P|\cdot)\right)$ (defined in the proof of Proposition 1) coincides exactly with the notion of the *inclusive divergence frontier*, as defined by Djolonga et al. [16]. It follows that the inclusive KL divergence frontier is related to the divergence curve we have defined as,

$$\mathcal{F}(P, Q) = \left\{ \left(c^{-1} \log t_1^{-1}, c^{-1} \log t_2^{-1}\right) \; : \; (t_1, t_2) \in \mathcal{C}(P, Q) \right\}\,.$$

### A.2 Generation Perplexity and Divergence Curves

Recall that the generation perplexity of a text distribution $P$ is the perplexity of this distribution under an external language model $R$. That is,

$$T_{\text{ppl}}(P) = \exp\left(-\mathbb{E}_P[\log R(\boldsymbol{x})]\right).$$

For simplicity, we write the perplexity using base $e$ rather than base 2. Then, the difference in generation perplexity between $P$ and $Q$ is given by

$$\left|T_{\text{ppl}}(P) - T_{\text{ppl}}(Q)\right| = \left|\exp\left(-\mathbb{E}_P[\log R(\boldsymbol{x})]\right) - \exp\left(-\mathbb{E}_Q[\log R(\boldsymbol{x})]\right)\right|$$

$$= \left|\exp\left(H(P) + \text{KL}(P|R)\right) - \exp\left(H(Q) + \text{KL}(Q|R)\right)\right|,$$

where $H(P) = -\mathbb{E}_P[\log P(\boldsymbol{x})]$ is the Shannon entropy of $P$. When $H(P) = H(Q) = \log C$, i.e., both $P$ and $Q$ are equally diverse, then

$$\left|T_{\text{ppl}}(P) - T_{\text{ppl}}(Q)\right| = C\left|\exp\left(\text{KL}(P|R)\right) - \exp\left(\text{KL}(Q|R)\right)\right|.$$

When $R = \lambda P + (1 - \lambda)Q$, this is proportional to the difference between the reciprocal of two coordinates of *one* point on the divergence curve. When $R$ is some other model, then $\left(\exp(-\text{KL}(Q|R)), \exp(-\text{KL}(P|R))\right)$ corresponds to the coordinates of a point enclosed within the divergence curve and the coordinate axes. Indeed, this is because the divergence curve encodes the Pareto frontier of $(\text{KL}(P|\cdot), \text{KL}(Q|\cdot))$.

When $H(P) \neq H(Q)$, the difference in the generation perplexity can be written as a function of some point $\left(\exp(-\text{KL}(Q|R)), \exp(-\text{KL}(P|R))\right)$ that is enclosed within the divergence curve and the axes:

$$\left|T_{\text{ppl}}(P) - T_{\text{ppl}}(Q)\right| = \left|C_1 \exp\left(\text{KL}(P|R)\right) - C_2 \exp\left(\text{KL}(Q|R)\right)\right|,$$

where $C_1 = \exp(H(P))$ and $C_2 = \exp(H(Q))$.

### A.3 Quantization: Definition and Illustration

We formally define the quantization of a distribution.

Consider a distribution $P$ over some space $\mathcal{X}$. Consider a partition $S = (S_1, \cdots, S_k)$ of $\mathcal{X}$, i.e., $\cup_{j=1}^{k} S_j = \mathcal{X}$ and $S_i \cap S_j = \varnothing$ if $i \neq j$. Quantizing the distribution $P$ over partitions $S$ gives us a multinomial distribution $\tilde{P}_S$ over $k$ elements. Concretely, we have,

$$\tilde{P}_S(j) = P(S_j).$$

This histogram is a classical example of a quantizer.

While the quantized distribution $\tilde{P}_S$ is a discrete multinomial distribution, it can be viewed as a piecewise constant approximation to $P$, similar to the histogram. This is visualized in Figure 3 for a two-dimensional example.

In our setting, $\mathcal{X}$ is the space of encoded representation of text, i.e., a Euclidean space $\mathbb{R}^d$. We use data-dependent quantization schemes such as $k$-means and lattice quantization of a learned feature representation.

In one-dimension, quantization is equivalent to computing a histogram. Hence, we casually use the term "bin" to refer to a partition.

### A.4 Pseudocode for MAUVE

Algorithm 1 shows the pseudocode for computing MAUVE. It consists of the following steps:

- The first step is to embed the sampled text using an external language model $M$. In our experiments, we use GPT-2 large [45].
- The second step is to quantize the embeddings. We primarily use $k$-means, which returns the cluster memberships $C_P$ and $C_Q$.

**Algorithm 1:** Pseduocode to compute MAUVE

**Input:** Human text $\{\boldsymbol{x}_i^P\}_{i=1}^N$, model text $\{\boldsymbol{x}_i^Q\}_{i=1}^{N'}$, number of clusters $k$, embedding model $M$, discretization $\Lambda$ of $[0,1]$.

**Output:** MAUVE$(P, Q)$.

`// Embed the samples`

$\{M(\boldsymbol{x}_i^P)\}_{i=1}^N, \{M(\boldsymbol{x}_i^Q)\}_{i=1}^{N'} \leftarrow$ `embed`$\left(M, \{\boldsymbol{x}_i^P\}_{i=1}^N, \{\boldsymbol{x}_i^Q\}_{i=1}^{N'}\right)$

`// Cluster embeddings jointly`

$C_P, C_Q =$ `quantize`$\left(\{M(\boldsymbol{x}_i^P)\}_{i=1}^N, \{M(\boldsymbol{x}_i^Q)\}_{i=1}^{N'}\right)$

`// Form quantized distributions by counting cluster assignments`

$\tilde{P} \leftarrow$ `count`$(C_P)/N, \ \tilde{Q} \leftarrow$ `count`$(C_Q)/N'$

`// Build the divergence curve`

Compute $\hat{\mathcal{C}}(\tilde{P}, \tilde{Q})$ from (4) for $\lambda \in \Lambda$

`// Compute` MAUVE `using numerical quadrature`

**return** $area\left(\hat{\mathcal{C}}(\tilde{P}, \tilde{Q})\right)$

---

- The third step is to form the quantized distributions from the cluster memberships from (3). This amounts to counting the number of points in each cluster contributed by $P$ and $Q$.

- The next step is to build the divergence curve. The full divergence curve (1) is a continuously parameterized curve for $\lambda \in (0, 1)$. For the sake of computation, we take a discretization $\Lambda$ of $[0, 1]$:

$$\hat{\mathcal{C}}(P, Q) = \left\{ \big( \exp(-c\,\mathrm{KL}(Q|R_\lambda)), \exp(-c\,\mathrm{KL}(P|R_\lambda)) \big) : \begin{array}{c} R_\lambda = \lambda P + (1-\lambda)Q, \\ \lambda \in \Lambda \end{array} \right\}. \tag{4}$$

  We take a uniform grid $\Lambda = \{1/n, 2/n, \cdots, (n-1)/n\}$ with $n$ points.

- The last step is to estimate the area under $\hat{\mathcal{C}}(\tilde{P}, \tilde{Q})$ using numerical quadrature.

## B   Software Package

We illustrate the use of the accompanying Python package, available on GitHub[8] and installable via pip[9] as `pip install mauve-text`.

Listing 1: Compute MAUVE from text

```
1  from mauve import compute_mauve
2
3  p_text = ... # list of strings representing human distribution P
4  q_text = ... # list of strings representing model distribution Q
5
6  # Obtain feature representation, quantize it and then compute MAUVE
7  out = compute_mauve(p_text=p_text, q_text=q_text,
8                      device_id=0,  # use GPU 0 for featurization
9                      max_text_length=256 # truncate text to 256 tokens
10                     )
11 print('MAUVE(P, Q) =', out.mauve)
12
13 # Plot the divergence curve
14 import matplotlib.pyplot as plt
15 plt.plot(out.divergence_curve[:, 0], out.divergence_curve[:, 1])
16
17 # Visualize quantized versions of P and Q
```

[8]https://github.com/krishnap25/mauve
[9]https://pypi.org/project/mauve-text/

```
18  import numpy as np
19  idxs = np.argsort(out.p_hist)[::-1]
20  sample_p = np.random.multinomial(n=1000, pvals=out.p_hist[idxs])
21  sample_q = np.random.multinomial(n=1000, pvals=out.q_hist[idxs])
22
23  x = np.arange(out.p_hist.shape[0])
24  plt.bar(x, sample_p, color='blue', alpha=0.3, label='P')
25  plt.bar(x, sample_q, color='red', alpha=0.3, label='Q')
26  plt.legend()
```

## C  Experiments: Setup

Here, we provide the full details of the experiments in §4. In particular, the outline of this appendix is as follows.

- Appendix C.1: the three task domains considered in the expeirments.
- Appendix C.2: training and decoding hyperparameters for each of these tasks.
- Appendix C.3: hyperparameters of MAUVE.
- Appendix C.4: details of other automatic comparison measures we consider.
- Appendix C.5: other details (software, hardware, running time, etc.).

### C.1  Task Domains

We consider an open-ended text generation task under three domains: web text, news and stories. As summarized in Table 2, we follow a slightly different setting for the task in each domain:

**Web text Generation.** The goal of this task is to generate articles from the publicly available analogue of the Webtext dataset[10] using pretrained GPT-2 models for various sizes. At generation time, we use as prompts the first 35 tokens of each of the 5000 articles from the Webtext test set, keeping maximum generation length to 1024 tokens (which corresponds, on average, to around 750 words). For comparison with human text, we use the corresponding human-written continuations from the test set (up to a maximum length of 1024 tokens).

**News Generation.** Under this task, the goal is to generate the body of a news article, given the title and metadata (publication domain, date, author names). We use a Transformer-based [56] causal language model, Grover [61], which is similar to GPT-2, but tailored to generating news by conditioning on the metadata of the article as well. Our generations rely on pretrained Grover architectures of various sizes. The generation prompt comprises the headline and metadata of 5000 randomly chosen articles from the April2019 set of the RealNews dataset [61], and the maximum article length was 1024 tokens. We reuse the publicly available Grover generations[11] for our evaluation.

**Story Continuation.** Given a situation and a (human-written) starting of the story as a prompt, the goal of this task is to continue the story. Here, we use a GPT-2 medium model fine-tuned for one epoch on the WritingPrompts dataset [18]. We use as generation prompts the first 50 tokens of 5000 randomly chosen samples of the test set of WritingPrompts. The machine generations are allowed to be up to 512 tokens long. The corresponding test examples, truncated at 512, tokens are used as human-written continuations.

### C.2  Training and Decoding Hyperparameters

We use size-based variants of Transformer language models [56] for training each task (domain). At decoding time, we explore a text continuation setting, conditioned on a prompt containing human-written text. All experiments were built using pretrained (and if applicable, finetuned) models implemented in the HuggingFace Transformers library [60]. The tasks are summarized in Table 2.

**Story Continuation Finetuning.** We finetune GPT-2 medium on the training set of the Writing-Prompts dataset using the cross entropy loss for one epoch over the training set with an effective

---

[10]https://github.com/openai/gpt-2-output-dataset
[11]available at https://github.com/rowanz/grover/tree/master/generation_examples

batch size of 32 and a block size of 512. We use the default optimizer and learning rate schedules of the HuggingFace Transformers library, i.e., the Adam optimizer with a learning rate of $5 \times 10^{-5}$.

**Decoding Hyperparameters.** We consider pure sampling (i.e., ancestral sampling from the model distribution), greedy decoding (i.e., choosing the argmax token recursively), and nucleus sampling [26] with parameter $p \in \{0.9, 0.92, 0.95, 0.99\}$ for web text generation and story continuation, and $p \in \{0.9, 0.92, 0.94, 0.96, 0.98\}$ for news generation.

## C.3   MAUVE Hyperparameters

MAUVE's hyperparameters are the scaling constant $c$, the embedding model $M$, and the quantization algorithm (including the size of the quantized distribution).

### C.3.1   Scaling Constant

Note that MAUVE's dependence on $c$ is order-preserving since the map $x \mapsto \exp(-cx)$ is strictly monotonic in $x$. That is, if $\text{MAUVE}_{c_1}(P, Q_1) > \text{MAUVE}_{c_1}(P, Q_2)$, then it holds that $\text{MAUVE}_{c_2}(P, Q_2) > \text{MAUVE}_{c_2}(P, Q_2)$ for all scaling constants $c_1, c_2 > 0$. In other words, the choice of the scaling constant affects the numerical value of MAUVE but leaves the relative ordering between different models unchanged. We choose $c = 5$ throughout because it allows for a meaning comparison between the numerical values of MAUVE; Appendix D.3 gives the values of MAUVE for various values of $c$.

### C.3.2   Embedding Model

We compute text embeddings from the GPT-2 large model. We find in Appendix D.3 that feature representations obtained from other large transformer models such as RoBERTA [34] also achieves similar results.

### C.3.3   Quantization

We experiment with three quantization algorithms.

**MAUVE-$k$-means.** We first run PCA on the data matrix obtained from concatenating the hidden state representations of the human text and model text. We keep 90% of the explained variance and normalize each datapoint to have unit $\ell_2$ norm. We then run $k$-means with FAISS for a maximum of 500 iterations for 5 repetitions; the repetition with the best objective value is used for the quantization. We quantize the human text distribution and the model text distribution by a histogram obtained from cluster memberships. We vary the number of clusters in $\{100, 250, 500, 1000\}$. Too few clusters makes the distributions seem closer than they actually are while too many clusters leads to many empty clusters (which makes all distributions seem equally far away). Yet, we find in Appendix D.3 that MAUVE with all these values of $k$ correlate strongly with each other; we use as default $k = 500$ clusters as it is neither too small nor too large.

**MAUVE-DRMM.** We use the code released by the authors of [22].[12] We take 10 components per layer and 3 layers for a total of 1000 components. We train the DRMM for 20 epochs using the hyperparameters suggested by the authors, i.e., a batch size of 64 with a learning rate

$$\gamma_t = \gamma_0 \min\{1, (2 - 2t/T)^2\},$$

where $T$ is the total number of updates and the initial learning $\gamma_0 = 0.005$. That is, the learning rate is set to a constant for the first half of the updates and then annealed quadratically. For more details, see [22, Appendix C].

**MAUVE-Lattice.** We use the code provided by the authors of [47].[13] We train a 4-dimensional feature representation of the hidden states for for 200 epochs using the triplet loss of [47], so that the learnt feature representations are nearly uniformly distributed. We use a 2-layer multilayer perceptron with batch normalization to learn a feature representation. We train this MLP for 200 epochs with hyperparameters suggested by the authors, i.e., a batch size of 64 and an initial learning rate of 0.1. The learning rate is cut to 0.05 after half the training and 0.01 after 75% of the training.

---

[12]https://github.com/PerttuHamalainen/DRMM
[13]https://github.com/facebookresearch/spreadingvectors

The learnt feature representations are then quantized using the lattice spherical quantizer into $744$ bins. This work as follows: let $S_r$ denote the integral points of the unit sphere of radius $r = \sqrt{50}$ in $\mathbb{R}^4$. A hidden state vector $x$ is run through the trained MLP $f$ to get its feature representation $f(x)$. Next, $f(x)$ is quantized to $\arg\min_{u \in S_r} \|f(x) - u/r\|_2^2$.

## C.4   Automatic Comparison Measures: Details and Hyperparameters

We now describe the other automatic comparison measures we compared MAUVE to, as well as their hyperparameters.

- **Generation Perplexity (Gen. PPL.)**: We compute the perplexity of the generated text under the GPT-2 large model.

- **Zipf Coefficient**: we report the slope of the best-fit line on log-log plot of a rank versus unigram frequency plot. Note that the Zipf coefficient only depends on unigram count statistics and is invariant to, for instance, permuting the generations. We use the publicly available implementation of [26].[14]

- **Repetition Frequency (Rep.)**: The fraction of generations which devolved into repetitions. Any generation which contains at least two contiguous copies of the same phrase of any length appearing at the end of a phrase is considered a repetition. We consider repetitions at the token level.

- **Distinct-$n$**: The fraction of distinct $n$-grams from all possible $n$-grams across all generations. We use $n = 4$.

- **Self-BLEU**: Self-BLEU is calculated by computing the BLEU score of each generations against all other generations as references. We report the Self-BLEU using $4$-grams. This operation is extremely expensive, so we follow the protocol of [26]: sample 1000 generations and compute the BLEU against all other 4999 generations. A lower Self-BLEU score implies higher diversity. This operation takes around 7 hours to compute on a single core of an Intel i9 chip (see hardware details in the next subsection).

- **Discriminator Accuracy**: We train a binary classifier to classify text as human or not. A smaller discrimination accuracy means that model text is harder to distinguish from human text. A separate classifier is trained for each model and decoding algorithm pair. For the story continuation task, we train a classification head on a frozen GPT-2 large model using the logistic loss. We use $25\%$ of the data as a test set and the rest for training; a regularization parameter is selected with 5-fold cross validation. For the news dataset, we follow the protocol of [61], i.e., a Grover mega model finetuned with a binary classification head. Results with other discriminators are reported in Appendix D.

## C.5   Miscellaneous Details

**Software.** We used Python 3.8, PyTorch 1.7 and HuggingFace Transformers 4.3.2.

**Hardware.** All the experiments requiring a GPU (finetuning, sampling generations and computing embeddings) were performed on a machine with 8 Nvidia Quadro RTX GPUs (24G memory each) running CUDA 10.1. Each only used one GPU at a time. On the other hand, non-GPU jobs such as computation of MAUVE and self-BLEU were run on a workstation with Intel i9 processor (clock speed: 2.80GHz) with 32 virtual cores and 126G of memory.

**Evaluation time for MAUVE.** Computation of MAUVE using $k$-means with 5000 generations takes $1 - 3$ minutes on *a single core* of an Intel i9 CPU (clock speed: 2.80GHz), using cached hidden state representations from a GPT-2 large (which are available during generation). On the other hand, MAUVE-DRMM takes 1.75 hours on a single CPU core while MAUVE-Lattice runs in about $5$ minutes on a single TITAN Xp GPU. MAUVE-$k$-means and MAUVE-DRMM can also run much faster on multiple CPU cores and can leverge GPUs although we did not use these features.

---

[14]https://github.com/ari-holtzman/degen/blob/master/metrics/zipf.py

| GPT-2 Size | Decoding | Gen. PPL | Zipf Coef. | Rep. | Distinct-4 | Self-BLEU | Human/BT($\uparrow$) | MAUVE ($\uparrow$) |
|---|---|---|---|---|---|---|---|---|
| small | Sampling | $101.880_{0.627}$ | $0.926_{0.001}$ | $0.001_{0.000}$ | $0.941_{0.001}$ | $0.327_{0.003}$ | $-27.52$ | $0.589_{0.018}$ |
|  | Greedy | $1.224$ | $1.037$ | $0.942$ | $0.072$ | $0.465_{0.000}$ | – | $0.008$ |
|  | Nucleus, 0.9 | $23.788_{0.144}$ | $1.012_{0.002}$ | $0.010_{0.001}$ | $0.859_{0.002}$ | $0.436_{0.004}$ | $-15.78$ | $0.878_{0.006}$ |
|  | Adversarial | $\mathbf{12.554}$ | $1.073$ | $0.006$ | $0.365$ | $0.525$ | – | $0.043$ |
| medium | Sampling | $129.263_{0.798}$ | $0.872_{0.001}$ | $0.001_{0.000}$ | $0.953_{0.001}$ | $0.281_{0.002}$ | $-30.77$ | $0.373_{0.010}$ |
|  | Greedy | $1.241$ | $0.978$ | $0.903$ | $0.091$ | $0.415$ | – | $0.012$ |
|  | Nucleus, 0.9 | $21.073_{0.134}$ | $\mathbf{0.957}_{0.001}$ | $0.005_{0.001}$ | $\mathbf{0.884}_{0.001}$ | $\mathbf{0.402}_{0.003}$ | $-3.43$ | $0.915_{0.006}$ |
|  | Adversarial | $\mathbf{12.554}$ | $1.006$ | $0.005$ | $0.381$ | $0.444$ | – | $0.044$ |
| large | Sampling | $30.080_{0.196}$ | $0.930_{0.002}$ | $\mathbf{0.002}_{0.001}$ | $0.916_{0.001}$ | $0.358_{0.001}$ | $-6.93$ | $0.845_{0.010}$ |
|  | Greedy | $1.232$ | $0.983$ | $0.881$ | $0.100$ | $0.413$ | – | $0.012$ |
|  | Nucleus, 0.95 | $13.499_{0.058}$ | $0.967_{0.002}$ | $0.006_{0.001}$ | $0.870_{0.001}$ | $0.412_{0.002}$ | $12.55$ | $0.936_{0.003}$ |
|  | Adversarial | $\mathbf{12.554}$ | $0.965$ | $0.005$ | $0.395$ | $0.429$ | – | $0.035$ |
| xl | Sampling | $31.886_{0.447}$ | $0.930_{0.001}$ | $0.002_{0.001}$ | $0.913_{0.001}$ | $0.360_{0.003}$ | $8.97$ | $0.882_{0.006}$ |
|  | Greedy | $1.278$ | $0.975$ | $0.859$ | $0.115$ | $0.417$ | – | $0.016$ |
|  | Nucleus, 0.95 | $14.143_{0.043}$ | $0.966_{0.002}$ | $0.005_{0.000}$ | $0.868_{0.001}$ | $0.413_{0.002}$ | $\mathbf{15.66}$ | $\mathbf{0.940}_{0.006}$ |
|  | Adversarial | $\mathbf{12.554}$ | $0.986$ | $0.005$ | $0.397$ | $0.448$ | – | $0.057$ |
| Human | n/a | $12.602$ | $0.952$ | $0.002$ | $0.878$ | $0.382$ | $47.25$ | – |

Table 6: Comparison measures across different model sizes, and decoding approaches for web text generations. Subscripts indicate the s.d. across 5 runs for the sampling-based methods; greedy decoding, being deterministic, always returns the same value for a given model. For nucleus sampling, we show the best hyperparameter value from $\{0.9, 0.92, 0.95, 0.99\}$ as per MAUVE. The column "Human/BT" gives the Bradley-Terry score obtained from a pairwise human evaluation (§4.3). Boldfaced numbers indicate best performance according to the measure, or closest to the human reference, when applicable. MAUVE shows that larger models perform better, across decoding approaches; moreover, nucleus sampling is the best decoding algorithm as per MAUVE.

| Grover Size | Decoding | Gen. PPL | Zipf Coef. | Rep. | Distinct-4 | Self-BLEU | % Disc. Acc.($\downarrow$) | MAUVE($\uparrow$) |
|---|---|---|---|---|---|---|---|---|
| base | Sampling | $37.505$ | $0.942$ | $0.002$ | $0.882$ | $0.419$ | $99.925$ | $0.700$ |
|  | Greedy | $1.413$ | $1.038$ | $0.518$ | $0.081$ | $0.548$ | $100.000$ | $0.005$ |
|  | Nucleus, 0.96 | $23.064$ | $0.974$ | $0.006$ | $\mathbf{0.847}$ | $0.462$ | $99.950$ | $0.701$ |
| large | Sampling | $27.796$ | $0.946$ | $\mathbf{0.002}$ | $0.878$ | $0.429$ | $99.450$ | $0.794$ |
|  | Greedy | $1.575$ | $1.012$ | $0.366$ | $0.124$ | $0.504$ | $100.000$ | $0.005$ |
|  | Nucleus, 0.98 | $20.792$ | $\mathbf{0.962}$ | $0.002$ | $0.859$ | $0.450$ | $98.475$ | $0.750$ |
| mega | Sampling | $22.656$ | $0.950$ | $0.001$ | $0.879$ | $0.427$ | $97.300$ | $0.808$ |
|  | Greedy | $1.796$ | $1.003$ | $0.316$ | $0.176$ | $0.500$ | $100.000$ | $0.005$ |
|  | Nucleus, 0.96 | $\mathbf{14.834}$ | $0.972$ | $0.003$ | $0.848$ | $\mathbf{0.469}$ | $\mathbf{88.675}$ | $\mathbf{0.813}$ |
| Human | n/a | $15.356$ | $0.956$ | $0.002$ | $0.842$ | $0.473$ | – | – |

Table 7: News generation evaluation across different Grover model sizes, and decoding approaches. For nucleus sampling, we show the best hyperparameter value from $\{0.9, 0.92, 0.94, 0.96, 0.98\}$ as per MAUVE. Disc. Acc. denotes the discrimination accuracy (%) of a Grover mega model trained to distinguish human text from machine text generated with the model and decoding algorithm of each row. Boldfaced numbers indicate performance closest to the human reference when applicable, or the best performance according to the measure. MAUVE favors nucleus sampling over ancestral sampling and greedy decoding.

# D  Experiments: Additional Results

We elaborate on the results in §4, including the results for the other domains. The outline is as follows.

- Appendix D.1: full results across model size and decoding (elaborating on §4.1).

- Appendix D.2: full results across text length (elaborating on §4.1).

- Appendix D.3: study of approximations in MAUVE (elaborating on §4.2).

- Appendix D.4: some miscellaneous plots such use of MAUVE for hyperparameter tuning.

Note that §4.3 is elaborated on in Appendix E.

| Decoding | Gen. PPL | Zipf Coef. | REP | Distinct-4 | Self-BLEU | % Disc. Acc. ($\downarrow$) | MAUVE($\uparrow$) |
|---|---|---|---|---|---|---|---|
| Sampling | $38.983_{0.143}$ | $\mathbf{1.066}_{0.002}$ | $\mathbf{0.001}_{0.000}$ | $0.833_{0.001}$ | $0.518_{0.003}$ | $0.781_{0.004}$ | $0.905_{0.010}$ |
| Nucleus, 0.9 | $15.433_{0.042}$ | $1.201_{0.002}$ | $0.006_{0.001}$ | $0.719_{0.001}$ | $0.637_{0.002}$ | $0.752_{0.004}$ | $0.887_{0.008}$ |
| Nucleus, 0.92 | $17.422_{0.060}$ | $1.179_{0.002}$ | $0.004_{0.001}$ | $0.742_{0.001}$ | $0.620_{0.003}$ | $0.720_{0.006}$ | $0.901_{0.005}$ |
| Nucleus, 0.95 | $\mathbf{21.599}_{0.127}$ | $1.147_{0.002}$ | $0.003_{0.000}$ | $\mathbf{0.775}_{0.002}$ | $0.589_{0.005}$ | $\mathbf{0.686}_{0.006}$ | $\mathbf{0.920}_{0.004}$ |
| Top-100 | $16.527_{0.041}$ | $1.252_{0.001}$ | $0.002_{0.000}$ | $0.743_{0.001}$ | $0.631_{0.001}$ | $0.782_{0.002}$ | $0.884_{0.007}$ |
| Top-500 | $23.833_{0.076}$ | $1.153_{0.001}$ | $0.001_{0.000}$ | $0.794_{0.001}$ | $\mathbf{0.576}_{0.002}$ | $0.697_{0.005}$ | $0.919_{0.005}$ |
| Greedy | $1.739$ | $1.362$ | $0.988$ | $0.101$ | $0.742$ | $0.997$ | $0.005$ |
| Human | $19.704$ | $1.101$ | $0.001$ | $0.783$ | $0.571$ | | |

Table 8: Story continuation evaluation across different and decoding approaches with GPT-2 medium. Disc. Acc. denotes the discrimination accuracy (%) of a classifier (a frozen GPT-2 large model with classification head) trained to distinguish human text from machine text generated with the decoding algorithm of each row. Boldfaced numbers indicate performance closest to the human reference when applicable, or the best performance according to the measure. MAUVE favors nucleus and top-$K$ sampling over ancestral sampling and greedy decoding.

| GPT-2 Size | Decoding | SP($\uparrow$) | JS($\downarrow$) | $\varepsilon$-PPL($\downarrow$) | Human/BT($\uparrow$) | MAUVE ($\uparrow$) |
|---|---|---|---|---|---|---|
| small | Greedy | 0.431 | 0.394 | 1049.589 | – | 0.008 |
| | Sampling | 0.653 | 0.425 | 19.401 | $-27.52$ | $0.589_{0.018}$ |
| | Nucleus, 0.9 | 0.652 | 0.414 | 25.938 | $-15.78$ | $0.878_{0.006}$ |
| medium | Greedy | 0.465 | 0.371 | 708.057 | – | 0.012 |
| | Sampling | 0.670 | 0.402 | 14.631 | $-30.77$ | $0.373_{0.010}$ |
| | Nucleus, 0.9 | 0.670 | 0.391 | 18.821 | $-3.43$ | $0.915_{0.006}$ |
| large | Greedy | 0.483 | 0.359 | 580.020 | – | 0.012 |
| | Sampling | 0.679 | 0.381 | 12.658 | $-6.93$ | $0.845_{0.010}$ |
| | Nucleus, 0.95 | 0.679 | 0.374 | 14.938 | 12.55 | $0.936_{0.003}$ |
| xl | Greedy | 0.496 | $\mathbf{0.349}$ | 497.696 | – | 0.016 |
| | Sampling | $\mathbf{0.686}$ | 0.369 | $\mathbf{11.412}$ | 8.97 | $0.882_{0.006}$ |
| | Nucleus, 0.95 | 0.686 | 0.363 | 13.677 | $\mathbf{15.66}$ | $\mathbf{0.940}_{0.006}$ |
| | Adversarial | n/a | n/a | n/a | – | 0.057 |

Table 9: MAUVE versus comparison measures based on language modeling (SP, JS and $\varepsilon$-PPL) across different model sizes, and decoding approaches for web text generations. SP, JS and $\varepsilon$-PPL are deterministic because they do not require generations from a decoding algorithm; moreover they cannot measure the quality of the adversarial decoding. The column "Human/BT" gives the Bradley-Terry score obtained from a pairwise human evaluation (§4.3). Boldfaced numbers indicate best performance according to the measure.

| Discriminator | BERT | | GPT-2 | | | Grover | | |
|---|---|---|---|---|---|---|---|---|
| | Base | Large | Small | Medium | Large | Base | Large | Mega |
| **Correlation** | 0.803 | 0.817 | 0.831 | 0.829 | 0.822 | 0.928 | 0.956 | 0.925 |

Table 10: Spearman rank correlation between the discrimination accuracy for various discriminators and MAUVE for news generation. All entries have a $p$-value of $< 2 \times 10^{-6}$.

| Decoding | Greedy | Beam $b = 4$ | Beam $b = 4$ + no 4-gram repeat | Beam $b = 8$ | Beam $b = 8$ + no 4-gram repeat | Ancestral | Nucleus |
|---|---|---|---|---|---|---|---|
| **Mauve** | 0.008 | 0.021 | 0.026 | 0.366 | 0.341 | $0.589_{0.02}$ | $\mathbf{0.878}_{0.007}$ |

Table 11: MAUVE and beam search: we compare beam search with beam sizes $b = 4, 8$ (with and without allowing 4-gram repetitions) with other decoding algorithms of Table 6 for web text generation with GPT-2 small. The subscript denotes the standard deviation over 5 random seeds, and is omitted for the deterministic greedy decoding and beam search.

## D.1 Comparison of Measures Across Model Size and Decoding

Full versions of Table 3 and Table 4 can be found between Table 6 for statistics-based measures and Table 9 for the language modeling measures. The corresponding tables for the news and story domains are Tables 7 and 8 respectively.

| GPT-2 size | Decoding | RoBERTa | GPT-2 |
|---|---|---|---|
| small | Sampling | 0.174 | 0.589 |
| | Greedy | 0.056 | 0.008 |
| | Nucleus, 0.9 | 0.723 | 0.878 |
| medium | Sampling | 0.292 | 0.372 |
| | Greedy | 0.114 | 0.011 |
| | Nucleus, 0.9 | 0.891 | 0.915 |
| large | Sampling | 0.684 | 0.845 |
| | Greedy | 0.125 | 0.012 |
| | Nucleus, 0.95 | 0.920 | 0.936 |
| xl | Sampling | 0.780 | 0.881 |
| | Greedy | 0.170 | 0.016 |
| | Nucleus, 0.95 | **0.947** | **0.940** |

Table 12: Comparison of MAUVE computed with dense embeddings from RoBERTa [34] large with the default GPT-2 large. Boldfaced numbers indicate best performance according to the measure. The two feature representations have a Spearman rank correlation of 0.993. See Figure 5 for a visual representation of a subset of this table.

**Note**: The main paper and the appendix treat the statistics-based measures differently (Gen. PPL., Zipf, Self-BLEU, etc). For each statistic $T$, the main paper (Tables 3 and 4) gives the difference $|T(Q) - T(P)|$ between the statistic on model text and human text, while in Tables 6, 7, 8 of the supplement, we show $T(Q)$ in the row corresponding to $Q$ and $T(P)$ in the row corresponding to human.

**Results.** From Table 6, we observe that among the decoding approaches, nucleus sampling achieves the best MAUVE followed by sampling and lastly by greedy decoding. This trend is consistent with the fraction of distinct 4-grams. On the other hand, in comparison with the perplexity of human text, Gen. PPL is too high for sampling and too low for greedy decoding; it does not give us a way to directly compare which of these two is better. MAUVE, however, rates greedy decoding as far worse than ancestral sampling. This is consistent with the empirical observation that greedy decoding produces extremely degenerate text [59]. Adversarial perplexity sampling produces unintelligible text which nevertheless has perfect Gen. PPL, thus demonstrating its unsuitability for as a comparison measure.

The results in Tables 7 and 8 for the news and story domains are qualitatively similar to the webtext domain. MAUVE, like discrimination accuracy, rates larger models as better and nucleus sampling as better than ancetral sampling and greedy decoding. An exception to this rule is Grover large, where MAUVE thinks ancestral sampling is better than nucleus sampling. The statistics-based measures Zipf coefficient, Repetition and the fraction of distinct 4-grams all prefer smaller Grover sizes.

Next we turn to the language modeling comparison measures in Table 9. JS consistently favors greedy decoding, which produces far worse text than other decoding algorithms. Likewise, $\varepsilon$-PPL favors ancestral sampling, which also produces somewhat degenerate text [26], while SP appears to be unable to distinguish between ancestral sampling and nucleus sampling. This makes SP, JS and $\varepsilon$-PPL unsuitable to compare generated text to human text.

While most measures behave nearly as expected across model architectures (larger models produce better generations for the same decoding algorithm), Self-BLEU prefers generations from GPT-2 medium over GPT-2 large or xl. This indicates that while measures based on word/token statistics are important diagnostic tools, they do not capture the quality of generated text entirely.

**Discriminator Accuracy: Choice of Discriminator.** We show the Spearman rank correlation between the discriminator accuracy for various choices of the discriminator in Table 10. The results show that MAUVE has a strong correlation with the discrimination accuracy for a variety of discriminators, including one based on a masked language model, BERT [14]. This correlation is particular strong for the Grover-based discriminators. We note that evaluating any one model and decoding algorithm pair requires fine-tuning a separate model. This can be particularly expensive for the larger models such as Grover mega. MAUVE, on the other hand, is inexpensive in comparison.

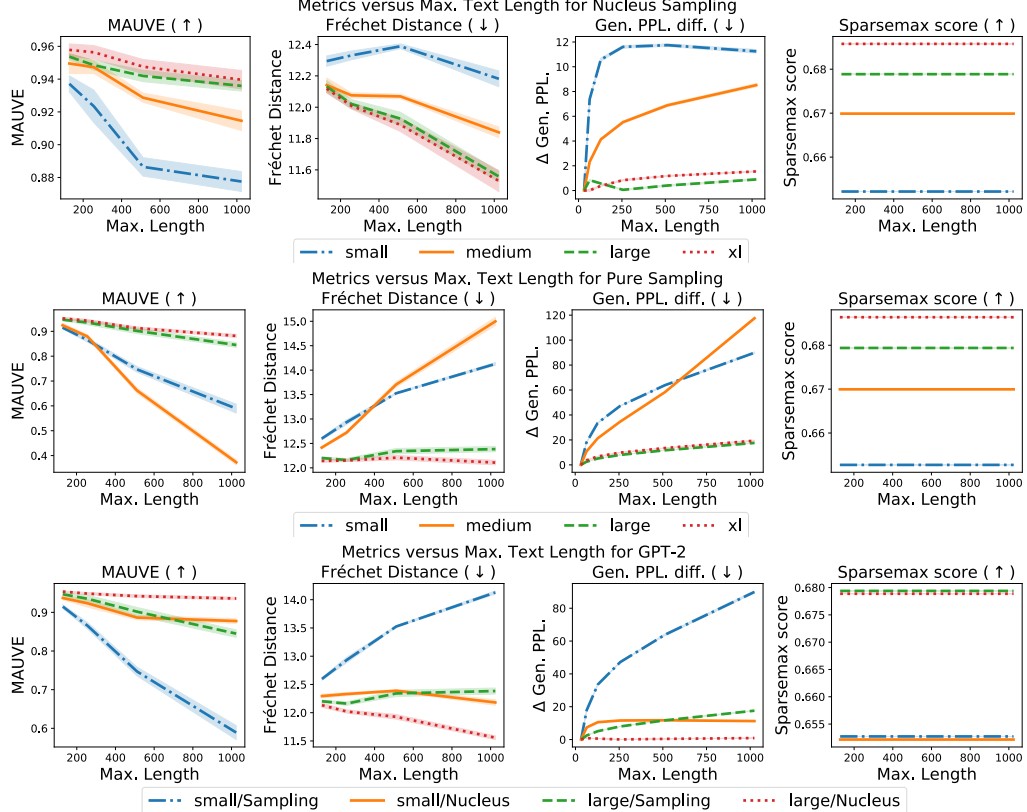

Figure 6: Generation quality versus maximum generation length as per various comparison measures for web text generation with GPT-2. We expect the quality of the generation to degrade as the maximum length of the text (both machine and human-written) increases. MAUVE is the only comparison measure which correctly shows this behavior across all models and decoding algorithms. The shaded area denotes one standard deviation over generations from 5 random seeds.

**Beam Search.** We also calculate MAUVE for beam search in Table 11. MAUVE is able to quantify the qualitative observations of Holtzman et al. [26]: beam search produces extremely degenerate text, but slightly better than greedy decoding. Disallowing repetition of 4-grams substantially improves the quality of the produced text, since the most glaring flaw of beam search is that the text is highly repetitive. However, the quality of the resulting text is still far worse than produced by ancestral sampling, and hence also nucleus sampling.

### D.2 Behavior Across Text Length

We now turn to the plot of comparison measures versus text length in Figure 6. We expect the quality of the generation to degrade as the maximum length of the text (both machine and human-written) increases.

**Comparison Measures.** Figure 6 plots MAUVE, Gen. PPL. and the Sparsemax score [39]. In addition we also plot the Fréchet distance, a variant of the Fréchet Inception Distance (FID) [25] which is the de facto standard evaluation metric for GANs in computer vision. The FID is computed as the Wasserstein-2 distance between Gaussians fit to the feature representation from using an Inception network; we adapt it to our setting by using embeddings from GPT-2 large instead. For Gen. PPL., we plot the difference of Gen. PPL., i.e., $|T_{\mathrm{ppl}}(Q_{\leq \ell}) - T_{\mathrm{ppl}}(P_{\leq \ell})|$, $T_{\mathrm{ppl}}(P_{\leq \ell})$ denotes the perplexity of the text $x \sim P$ truncated at a length of $\ell$. The perplexity is measured using GPT-2 large model as the external language model.

**Results.** MAUVE indeed shows this expected behavior. However, the Fréchet distance [25] actually decreases for nucleus sampling for all GPT-2 sizes and ancestral sampling for GPT-2 xl. This shows

that it is not suitable as an evaluation metric for text. While Gen. PPL. mostly agrees with MAUVE about quality versus text length, we observe non-monotonic behavior for nucleus sampling with GPT-2 small and large. Finally, sparsemax score [39] does not depend on the samples generated and is therefore independent of the maximum text length.

### D.3 Effect of Approximations of MAUVE

We expand upon the approximation results from the main paper in §4.2.

**Embedding Model.** Table 12 shows MAUVE compute with RoBERTa large in addition to the default GPT-2 large. We restrict the maximum text length of the RoBERTa model to 256 BPE tokens, since RoBERTa cannot handle sequences of length 1024 tokens. We observe similar trends with both: larger models are rated higher and nucleus sampling is preferred over ancestral sampling while greedy decoding is rated very low. The Spearman rank correlation between MAUVE computed with the two feature representations is 0.993, indicating that MAUVE is robust to feature representations. We observe that RoBERTa penalizes ancestral sampling more while rating greedy decoding higher across all model sizes. We leave a study of the biases induced by different feature representations to future work.

**Quantization Algorithm.** We compare different choices of the quantization to $k$-means with $k = 500$, which is our default. The Spearman rank correlation between MAUVE computed with $k$-means for $k$ ranging from 100 to 5000 correlates nearly perfectly with that of $k = 500$. In particular, the Spearman correlation is exactly 0.99 or 1.00. Likewise, MAUVE computed with DRMM or lattice quantization has a near-perfect Spearman correlation of at least 0.99 with $k$-means. While the actual numerical value of MAUVE could vary with the quantization algorithm, these results show that the *rankings induced by various variants of* MAUVE *are nearly identical*.

See Figure 8 (Left) for how MAUVE-$k$-means depends on the number of clusters, $k$. If $k$ is too small ($k < 100$), all methods are scored close to 1. If $k$ is too large $k > 2000$), all methods are scored close to 0. There is a large region between these two extremes where MAUVE-$k$-means is effective.

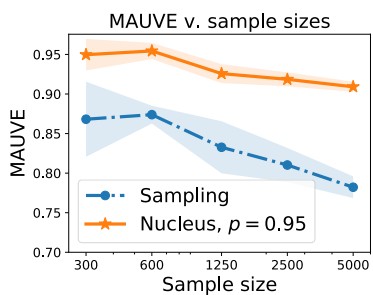

Figure 7: Effect of the sample size on MAUVE.

**Effect of Number of Generations.** Figure 7 plots the value of MAUVE versus the sample size $n$, with the number of clusters in $k$-means chosen as $k = n/10$. We observe that a smaller sample size gives an optimistic estimate of MAUVE; this is consistent with [16, Prop. 8]. We also note that a smaller sample size leads to a larger variance in MAUVE.

### D.4 Miscellaneous Plots

Figure 8 plots MAUVE for nucleus and top-$K$ sampling for various values of the hyperparameters $p$ and $K$.

## E Human Evaluation: Protocol and Full Results

Here, we describe the human evaluation protocol and results of §4.3 in detail. The outline for this section is

- Section E.1: Overview of the human evaluation setup.
- Section E.2: Details of the statistical model we fit to the raw data.
- Section E.3: Full results of the human evaluation.
- Section E.4: Additional details of the human evaluation protocol.

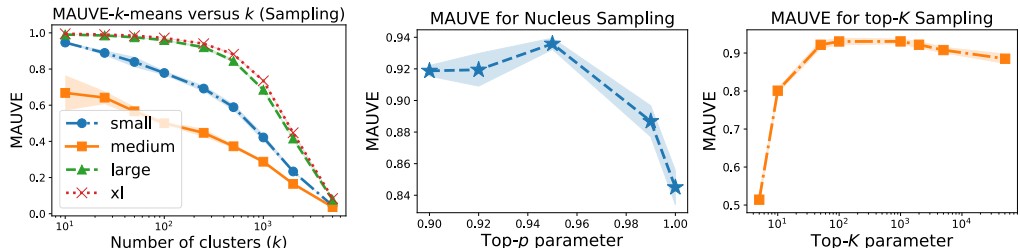

Figure 8: **Left**: MAUVE-$k$-means for various values of the number of clusters $k$. We use $k = 500$ as our default because it is neither too small (every method is scored close to 1) nor too large (every method is scored close to 0). **Center & Right**: MAUVE for nucleus and top-$K$ sampling for different values of $p$ and $K$ for GPT-2 large. MAUVE rates nucleus sampling with $p = 0.95$ and top-$K$ sampling with $100 \leq K \leq 1000$ as the best choices. The shaded area denotes one s.d. over generations from 5 random seeds.

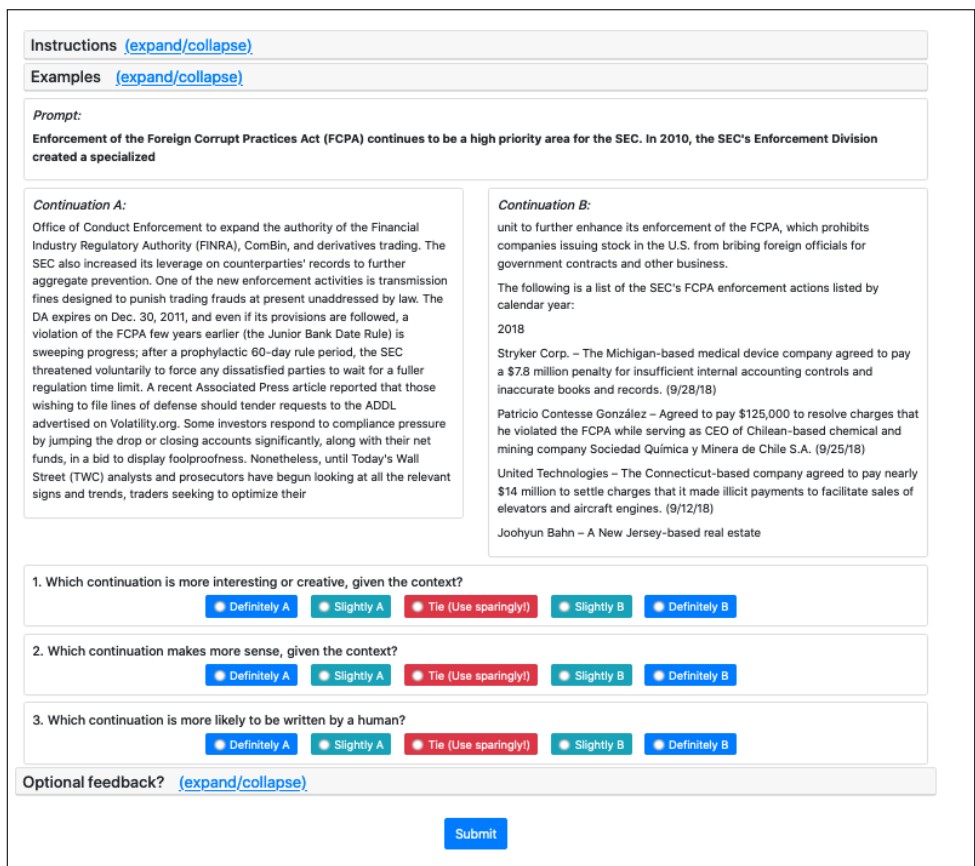

Figure 9: Mechanical Turk interface for human evaluation.

## E.1 Overview

We performed a human evaluation for web text generations where human annotators are instructed to select one from a pair of texts. The pairs might come from human and machine text, or different sources of machine text; each is based on the same prompt for generation (recall that we obtained the prompt as a prefix from the human text).

The annotators were presented with a pairs of continuations of the same prompt and were instructed to choose which one is (a) more interesting, (b) more sensible, and, (c) more likely to be written by a human. Each question could have a different answer.

We considered all four GPT-2 model sizes with pure sampling and nucleus sampling. We collected 90 annotations for each of the 8 model-human pairs and $\binom{8}{2}$ model-model pairs on the Amazon Mechanical Turk platform using the interface shown in Figure 9. We fit a Bradley-Terry model to obtain a ranking from the pairwise preferences of the crowd-workers. We report the correlation of MAUVE with obtained Bradley-Terry scores.

### E.2 From Pairwise Preferences to Ranking: the Bradley-Terry Model

We compute the Bradley-Terry (BT) scores from the pairwise preferences obtained from the human evaluation along each of the three axes interesting, sensible and more likely to be written by a human.

**Bradley-Terry Model Review.** Given $n$ players with scores $w_1, \cdots, w_n$, the the Bradley-Terry model [37] models the outcome of a head-to-head comparison of any two players using a sigmoid[15]

$$\text{Prob}(i \text{ beats } j) = \frac{1}{1 + e^{-(w_i - w_j)/100}} .$$

The model also assumes the outcome of each head-to-head comparison of any pair of players is independent of all other comparisons. Note that the model is invariant to additive shifts of the scores, i.e., the model probabilities induced by scores $w_1 + C, \cdots, w_n + C$ is same as the that induced by $w_1, \cdots, w_n$ for any constant $C$. For uniqueness, we normalize the scores so that their mean is 0.

**Fitting the Model.** The Bradley-Terry model can be fit to data using Zermelo's algorithm [27]. Suppose that we are given a dataset of head-to-head comparisons summarized by numbers $N_{ij}$ denoting the number of times player $i$ has defeated player $j$. Then, the negative log-likelihood $\ell(w_1, \cdots w_n)$ of the data under the Bradley-Terry model can be written as

$$\ell(w_1, \cdots, w_n) = -\sum_{i=1}^{n} \sum_{j=1}^{n} N_{ij} \log(1 + e^{-(w_i - w_j)/100}) .$$

This is convex in the parameters $w_1, \cdots, w_n$ since the log-sum-exp function is convex. Zermelo's algorithm [27] can be used to compute the maximum likelihood estimate. Denote $\widetilde{w}_i = w_i/100$. Starting from an initial estimate $\widetilde{w}_1^{(0)}, \cdots, \widetilde{w}_n^{(0)}$, each iteration of Zermelo's algorithm performs the update

$$u_i^{(t)} = \log\left(\sum_{j \neq i} N_{ij}\right) - \log\left(\sum_{j \neq i} \frac{N_{ij} + N_{ji}}{\exp(\widetilde{w}_i^{(t)}) + \exp(\widetilde{w}_j^{(t)})}\right)$$

followed by the mean normalization

$$\widetilde{w}_i^{(t+1)} = u_i^{(t)} - \frac{1}{n} \sum_{j=1}^{n} u_j^{(t)} .$$

**Processing Raw Data.** We collect the result of a head-to-head comparison using 5 options: Definitely A/B, Slightly A/B or a Tie. We combine Definitely A and Slightly A into a single category denoting that A wins, while ties were assigned to either A or B uniformly at random.

### E.3 Full Results of the Human Evaluation

**BT Model for Human Eval.** In our setting, each "player" is a source of text, i.e., one human, plus, eight model and decoding algorithm pairs (four model sizes GPT-2 small/medium/large/xl coupled with pure sampling or nucleus sampling). We compute the BT score of each player as the maximum likelihood estimate of corresponding the parameters $w_1, \cdots, w_n$ based on head-to-head human evaluation data.

A higher BT score indicate a stronger preference from human annotators. The BT scores are reported in Table 13. The Spearman rank correlations between each of these scores are ($p$-value $\leq 5 \times 10^{-4}$ for each):

---
[15]the scaling factor 100 is arbitrary and does not change the model

|  |  | BT/Human-like | BT/Interesting | BT/Sensible |
|---|---|---|---|---|
| Human |  | 47.251 | 25.503 | 43.229 |
| xl | Nucleus, $p = 0.95$ | **15.664** | **23.046** | **31.888** |
|  | Sampling | 8.966 | 9.529 | 7.753 |
| large | Nucleus, $p = 0.95$ | 12.553 | 6.785 | 8.781 |
|  | Sampling | $-6.935$ | $-1.532$ | $-7.106$ |
| medium | Nucleus, $p = 0.9$ | $-3.429$ | $-12.824$ | $-7.293$ |
|  | Sampling | $-30.769$ | $-34.323$ | $-32.004$ |
| small | Nucleus, $p = 0.9$ | $-15.783$ | $-0.697$ | $-7.442$ |
|  | Sampling | $-27.518$ | $-15.487$ | $-37.805$ |

Table 13: Fitted Bradley-Terry (BT) scores for each of the three axes rated by human annotators: "Human-like" denotes measures how likely the text is to be written by a human, while "Interesting" and "Sensible" quantify how interesting or sensible the text is. The Spearman rank correlations between each of these scores are ($p$-value $\leq 5 \times 10^{-4}$ for each): Human-like and Interesting: 0.917, Human-like and Sensible: 0.917, Interesting and Sensible: 0.967.

|  | Gen. PPL | Zipf Coef. | REP | Distinct-4 | Self-BLEU | MAUVE |
|---|---|---|---|---|---|---|
| BT/Human-like | 0.810 | 0.833 | $-0.167$ | 0.738 | 0.595 | **0.952** |
| BT/Interesting | 0.643 | 0.524 | $-0.143$ | 0.524 | 0.405 | **0.810** |
| BT/Sensible | 0.738 | 0.690 | $-0.071$ | 0.595 | 0.524 | **0.857** |

Table 14: Spearman rank correlation between the Bradley-Terry scores from the human evaluation and the various automatic comparison measures.

- Human-like and Interesting: 0.917,
- Human-like and Sensible: 0.917,
- Interesting and Sensible: 0.967.

**Interpreting BT scores.** The BT scores reported in Table 13 give us predictions from the sigmoid model above. For example, consider the column "BT/Human-like". The best model-generated text, GPT-2 xl with nucleus sampling, will lose to human text with probability 0.578. At the other end, GPT-2 small with nucleus sampling will lose to human text with probability 0.679. This shows that there is still much room for improvement in machine generated text.

**Discussion.** In general, the BT scores from human evaluations and MAUVE both indicate that (a) nucleus sampling is better than pure sampling for the same model size, and, (b) larger model sizes are better for the same decoding algorithm. There is one exceptions to this rule, as per both the human evaluations and MAUVE: GPT-2 small is better than GPT-2 medium for pure sampling.

**Correlation Between Comparison Measures.** We compare the Spearman rank correlation between the various automatic comparison measures and the BT scores from human evaluations in Table 14. In terms of being human-like, we observe that MAUVE correlates the best (0.95) with human evaluations. While this is also the case for Zipf coefficient, we note that it is based purely on unigram statistics; it is invariant to the permutation of tokens, which makes it unsuitable to evaluate generations.

We note that MAUVE does disagree with human evaluations on specific comparisons. For instance, MAUVE rates nucleus sampling with GPT-2 medium as being better than pure sampling from GPT-2 large and xl. The same is also the case with Gen. PPL. We leave a detailed study of this phenomenon to future work.

### E.4 Additional Details

We describe more details for the human evaluation. The terminology below is taken from [54].

**Number of Outputs Evaluated.** We compare 9 players: one player is "human", representing human-written text, whereas the other 8 are text generated by the model using the first 35 tokens of the corresponding human generation as a prompt. Each of the 8 non-human players come from a GPT-2 model of different sizes (small, medium, large, xl) and two decoding algorithms (pure sampling

and nucleus sampling). We perform 90 comparisons between each pair of players, so each player is evaluated $90 \times 8 = 720$ times.

**Prompt Filtering.** We manually selected 1831 out of 5000 prompts which are well-formed English sentences from the webtext test set[16]. For every head-to-head comparison, we sample 90 prompt without replacement and then sample the corresponding completions (for human-generated text, we use the test set of webtext). We only consider a pair of players for human evaluation if the generation from each player is at least 200 BPE tokens long (and we truncate each generation at a maximum length of 256 BPE tokens).

**Number of Evaluators.** 214 unique evaluators participated in the evaluation. Of these, 11 evaluators supplied at least 50 annotations 95 evaluators supplied at least 10 annotations.

**Evaluator Selection and Pay.** We conduct our human evaluation on Amazon Mechanical Turk. Since the task only requires elementary reading and understanding skills in English, we open the evaluations to non-experts. Each crowd-worker was paid 0.40 per annotation. The pay was estimated based on a \$16/hour wage for the $85^{\text{th}}$ percentile of response times from a pilot study (which was approx. 98 seconds per annotation). There evaluators are not previously known to the authors.

**Training and Instructions.** The evaluators were given instructions about the task and two detailed examples. No other training was provided due to the elementary nature of the task. The screenshots of these examples are given in Figure 10 while the instructions read:

> **Task Info**: We are studying how good AI models are at generating text on the internet. You are given a snippet of text from a random document on the internet, called the "prompt" or the "context", as well as and two continuations, A and B. One or both of these is written by an AI. You must choose (a) which of two continuations is more interesting, (b) which makes more sense given the prompt, and, (c) which is more likely to have been written by a human, as per your assessment.
>
> **Guidelines**:
>
> - There are five choices for each question: Definitely A/B, Slightly A/B, or Tie. Please use the "Tie" option extremely sparingly! (No more than one in every ten pairs should be chosen as a tie along any of the three questions).
> - The questions can have different answers! Some text is very creative or interesting, but it doesn't quite fit the prompt or make sense.
> - Try to focus on quality over quantity. The text can be long but contain rambly gibberish.
> - Don't worry if the text ends abruptly, or has other artifacts of the website downloading process (text like 'Advertisement' for instance).
> - Please do your best, some of these are pretty challenging!
> - Answering each question should take around 1.5 minutes on average, as per our estimation. We have calibrated the pay to be \$16 per hour with this speed.

**Quality Control.** All annotations made in under 25 seconds were excluded for quality control (the mean response time per annotation was 47 seconds).

**Quality Criteria.** We use three quality criteria. The questions asked to the evaluators are (verbatim):

1. Interestingness: "Which continuation is more interesting or creative, given the context?"
2. Sensible: "Which continuation makes more sense, given the context?"
3. Human-like: "Which continuation is more likely to be written by a human?"

Note that we do explicitly name the criteria in the evaluation form, although those names could be inferred from the definitions. We use these names only in the paper.

Further Details:

- Each of the criteria is a "Goodness" criteria as per the classification of [3]. Goodness refers to the setting where there is no single, general mechanism for deciding when outputs are maximally good, only for deciding for two outputs which is better and which is worse. E.g. for Fluency, even

---

[16]The webtext dataset is scraped from the internet and is *not* curated. It contains poor prompts such as headers of webpages or error message, such as: "Having trouble viewing the video? Try disabling any ad blocking extensions currently running on your browser" or "Front Page Torrents Favorites My Home My Galleries Toplists Bounties News Forums Wiki". We exclude such prompts as they are unsuitable for human evaluation.

if outputs contain no disfluencies, there may be other ways in which any given output could be more fluent.

- Each criterion assesses outputs as a whole, not just form or just content.
- The output quality is assessed without referring to anything other than the output itself, i.e. no system-internal or external frame of reference.
- Each criterion involves a subjective assessments of preferences by evaluators.
- The quality of outputs is assessed *without* considering their *effect* on something external to the system, e.g. the performance of an embedding system or of a user at a task.
- For each criteria, we provide 5 options: "Definitely/Slightly A/B" and "Tie (Use sparingly!)"

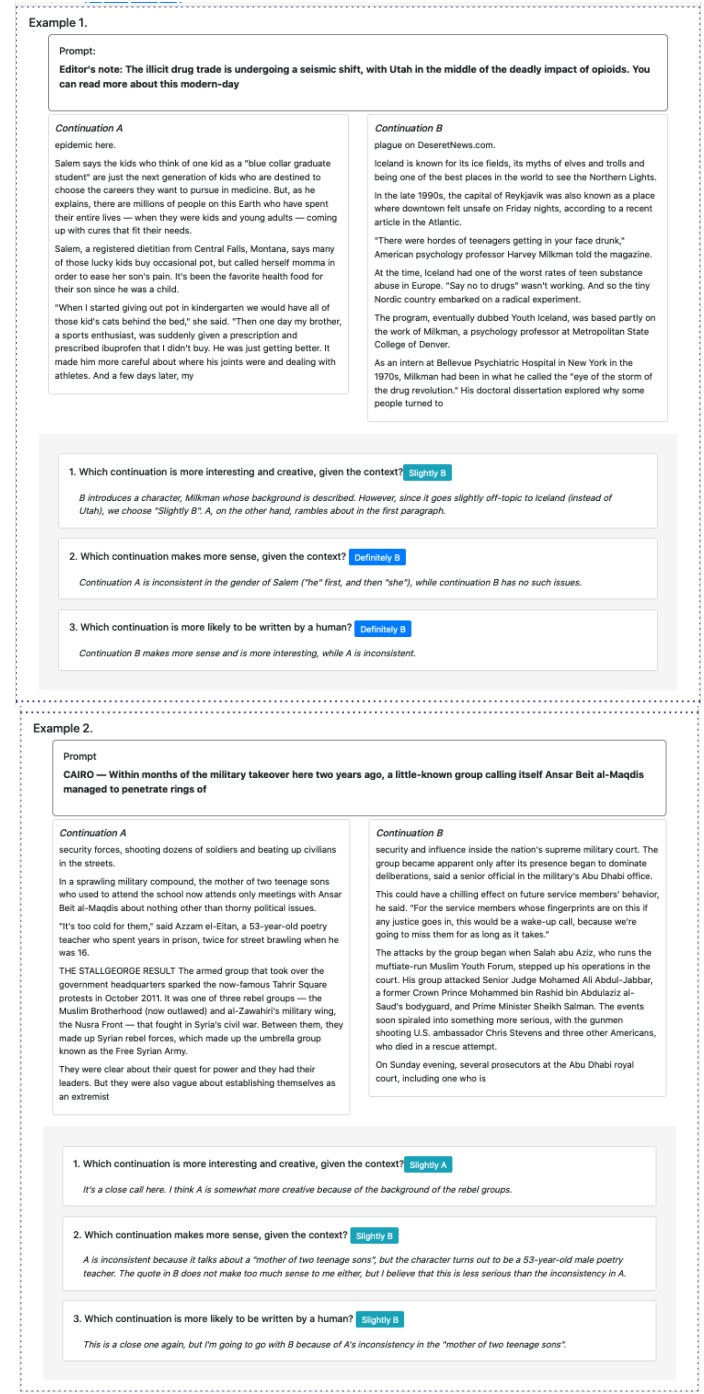

Figure 10: Annotated examples shown to the evaluators.

## F   Interpreting the Quantization

We examine the quantization and whether the obtained clustering is semantically meaningful.

We consider the news domain because the prompts from the RealNews datatset [61] also contain some metadata not used by MAUVE. We examine the *domain* of the generations in each cluster, which refers to the website from which the article was downloaded, e.g., *nytimes.com*. There are a total of 150 domains in the data. We analyze the cluster memberships calculated during the computation of MAUVE$(P, Q)$, where $P$ is the human distribution and $Q$ refers to Grover Mega with nucleus sampling ($p = 0.96$) and the number of clusters is $k = 500$.

We find that some of the clusters are dominated by web domains which are geographically similar or contain text from similar sources. In particular, of the 21 clusters which had at least 20 samples each, we find that:

- 7 clusters contain exactly one or two web domains each;
- Cluster 254 comprised web domains from Australia: *bordermail.com.au*, *dailyadvertiser.com.au*, *theherald.com.au*
- Cluster 51 comprised of web domains from Canada, namely *calgaryherald.com*, *canada.com*, *edmontonjournal.com*, *montrealgazette.com*, *ottawacitizen.com*, *theprovince.com*, *torontosun.com*, *vancouversun.com*. It also contains one outlier from Baltimore, USA: *baltimoresun.com*;
- Cluster 391 comprised 8 web domains from the UK: *bbc.com*, *bournemouthecho.co.uk*, *heraldscotland.com*, *theguardian.com*, *thenorthernecho.co.uk*, *capitalfm.com*, *thecourier.co.uk*, *dailymail.co.uk*, *pressandjournal.co.uk*;
- Cluster 322 contains domains from South Asia: *thedailystar.net*, *mangalorean.com*, *indianexpress.com*, *nation.com.pk*, *thenews.com.pk*, *dailytimes.com.pk*, *dawn.com*, as well as one outlier *bbc.com*;
- Cluster 251 contained only web domains owned by the NBC network: *nbcdfw.com*, *nbcmiami.com*, *nbcphiladelphia.com*, *necn.com*
- Cluster 421 only contained 3 financial/investment web domains: *etftrends.com*, *fool.com*, *seekingalpha.com*.

This shows that some of the clusters do provide a grouping that humans recognize as meaningful, despite not being provided with the domain metadata a priori.

## G   Example Generations

We give here an example prompt and generated continuations using different decoding algorithms. Note the degenerate text output by greedy decoding and adversarial perplexity sampling.

**Prompt.**

> Several people have asked about the techniques we used when cleaning out my mom's fabric stash last week. There are of course many ways to purge, sort, and organize fabric,

**Pure Sampling.**

> Several people have asked about the techniques we used when cleaning out my mom's fabric stash last week. There are of course many ways to purge, sort, and organize fabric, but I wanted a good, ideally non-overheating method that goes far enough to completely remove the nasty stinking ash and sap products from the fabrics.
>
> So, this post describes some of the methods I researched and used. Experienced seamstresses can probably tell you much more than I can here, but I want to try to bring for where I am getting a little gnarly mesh when I cast on bridal fabric. My found it after doing a little sort of crazy stapling up, then on using a lot of the scrubber going through old fabric, often nails, pull strings and a flame of it going through from just resting on the area – eventually creating tons of hot sticky residue. So, porch nails, the like may help. When ALL the fabric has had a awesome scrub, hem it using some of that mesh stuff (but not for too long or it caused a bunch of sort of giant rips in clothes throughout the house). Bagged out the dog

food, settlers, weeded out a lot of fine mesh that stuck, and then folded this mesh back up – then placed it in a zippered bag and packed it up to place in my closet, bag or chest if it was from Panda Proof revival. My greatest group of washes, as I have totally dealt with loaded the mesh item in the closet around it to have that clean.

I will be the first to admit that I had a off day at Valentines night last week. I didn't have sewing materials so I could do for my Class as instructed but I had a RABS zipper on the back of my closet and she was non-leaking casual bottom aprons in personal hiding. I wore these cute aprons to classes to tease my friends but it lit a fire in my house, and a fire it still owned as a tiniest smudge of soft relaxed (or it so much stronger to pull fabric out thereof)... Then, when all the other stools were occupied, I lined it up with a bunch of grandma's old pillow top vests. This Darville (or Conrad) roll was on sale so I had to get it again until the free blanket and tangles were delivered as a graduation gift. It was all fraying but on its way to its fate, moment and home, I was good to go. Renovations to the bars are underway :-)

Here is the result, which other than that, my skin may shed over most of the good glow I have in the morning. They make beautiful skidoos. Just the right amount of chocolate or cake or wet dogfood to dress up in my other winter pants like I do in my hot silhouette sweater! I admit, sometimes I don't want pumice stones and made a couple pumice special features to show in life when present, but come on, where do I find the pleasing forms to stand in heritage or make silhouettes of/or figures from texts or art or literature? Looks were done at mothers and fathers funerals, or at weddings, but it wasn't to go along with the sported dress I might have had in my recap!

## Nucleus Sampling, $p = 0.95$.

Several people have asked about the techniques we used when cleaning out my mom's fabric stash last week. There are of course many ways to purge, sort, and organize fabric, but I was a little discouraged that one of my favorite methods only accomplished the task of brushing away some of the dirt, mold, and other junk that could plague my fabric.

Within a few days of my clean-out, I had a great deal of décor management and was able to stash the excess items away for safekeeping.

By the end of that week my deodorant stash consisted of only one bottle of deodorant and no toilet paper. As you can imagine I was buzzing with delight.

Mark, our creative chef, has a code of ethics for all things edible. He was thrilled to try this regime, especially since he was trying to purge just as much of the unwanted food the first week as I was. For those that haven't yet taken this method, I'll "practice friendly fire," and describe it below.

1. Tape Aside All Contraband Trays

I don't know how many is more embarrassing for you to have accessible and unexplained food, junk, germs, etc. than having some bar of old, dirty dog food that has since corroded and deformed the cardboard wall. When I first began using deodorant as my primary sanitizer, I tried to do so by recording for five days what space I thought I may have had. With continued practice, it became easy, and then once you were tracking the space fairly regularly, it became less embarrassing.

Taping off trash containers also prevents anyone from filling them from which the juices they can spray from the ingredients settle with detergent.

For foamy can liners, attach bar tape along the top of the can. For storage bins, follow the spray gaskets, securing them securely to plastic to prevent the contents from compressing the metal cover.

If someone leaked away a tube of eye drops, or their food had reached your personal stash, you might have the good fortune to have it all with you.

2. Leave Cooler Volumes Empty In Your Pantry

You cannot truly sanitize a fresh batch of eggs and yogurt, let alone a box of 1/2 gallon freezer bins. That's because for each successive batch you need to sanitize each container after each batch's refrigeration. Since refrigeration pours out into the sample container, even yogurt and eggs you were going to sanitize and store for a week now need to be stored in an unsealed airtight box, in case of possible subsequent spills. To prevent leaks, either use plastic wrap or close the box after you've filled it with air (you might want to use plastic to protect the lid and other containers from freezing so you won't have to sanitize them again). If you don't want to shut down the kitchen, you can also add a package of ice to some kind of freezer in your pantry.

It's worth noting that for eggs and yogurt, and other specialty foods, your freezer needs to be no more than 6-8 hours old, so you don't want to pack lids that need an extra week to

guarantee an expiration date. Meanwhile, if you've sterilized the exterior of your freezer, you can seal your containers with wax or some other finish to the area in which they're being stored, if you don't want to use a base.

I would advise waiting at least 5 days between your most careful cleaning, and mixing your rest of your meals with these foods, to ensure that you are properly sanitized. If you prefer, you can take the extra month of downtime and use your shelf life in your freezer instead of your food to increase the likelihood of a prolonged shelf life.

3. Swap Primary Sanitizers

I cannot emphasize how important it is for primary chemicals to be readily available when you use them for cleansing. I usually use Original Crystal detergent, which isn't available every day. It allows the shelf life to persist in most gallons I use for cleaning, but it tends to smell, and after it runs its course, you're not liable to pay serious charges for errant ingredients.

You might also need water, or some hot water. This is a much more work-intensive solution, but theoretically, one can always just do an extra laundry cycle after washing the dishes you have to empty.

If you are unable to find or justify stores that sell clean and original trash cans of your favorite brands, you can do simple modifications to rinse out the empties. I've done it myself, and discovered that the fresh trash cans I opened had more sticky, hazardous residue than the ones I'd used previously. Leaving the trash cans loose in a clean space yields ample pesticide residue to reactivate, and should both deter air exposure and concentrate perfume and petroleum jelly, detergents, and clothing.

## Greedy Decoding.

Several people have asked about the techniques we used when cleaning out my mom's fabric stash last week. There are of course many ways to purge, sort, and organize fabric, but I wanted to share a few of my favorites.

1. The "Dirty" Method

This is the most basic method. It's the one I use most often. It's also the one I'm most proud of.

First, you need to find a place to put your fabric. I like to put it in a corner of my closet, but you can use any corner.

Next, you need to get a small, sharp knife. I like to use a small, sharp knife. I like to use a small, sharp knife.

Next, you need to cut a small hole in the fabric. I like to use a small, sharp knife. I like to use a small, sharp knife.

Next, you need to put the fabric in the hole. I like to use a small, sharp knife. I like to use a small, sharp knife.

Next, you need to put the fabric in the hole. I like to use a small, sharp knife. I like to use a small, sharp knife.

Next, you need to put the fabric in the hole. I like to use a small, sharp knife. I like to use a small, sharp knife.

Next, you need to put the fabric in the hole. I like to use a small, sharp knife. I like to use a small, sharp knife.

Next, you need to put the fabric in the hole. I like to use a small, sharp knife. I like to use a small, sharp knife.

Next, you need to put the fabric in the hole. I like to use a small, sharp knife. I like to use a small, sharp knife.

Next, you need to put the fabric in the hole. I like to use a small, sharp knife. I like to use a small, sharp knife.

Next, you need to put the fabric in the hole. I like to use a small, sharp knife. I like to use a small, sharp knife.

Next, you need to put the fabric in the hole. I like to use a small, sharp knife. I like to use a small, sharp knife.

Next, you need to put the fabric in the hole. I like to use a small, sharp knife. I like to use a small, sharp knife.

Next, you need to put the fabric in the hole. I like to use a small, sharp knife. I like to use a small, sharp knife.

Next, you need to put the fabric in the hole. I like to use a small, sharp knife. I like to use a small, sharp knife.

Next, you need to put the fabric in the hole. I like to use a small, sharp knife. I like to use a small, sharp knife.

Next, you need to put the fabric in the hole. I like to use a small, sharp knife. I like to use a small, sharp knife.

Next, you need to put the fabric in the hole. I like to use a small, sharp knife. I like to use a small, sharp knife.

Next, you need to put the fabric in the hole. I like to use a small, sharp knife. I like to use a small, sharp knife.

Next, you need to put the fabric in the hole. I like to use a small, sharp knife. I like to use a small, sharp knife.

Next, you need to put the fabric in the hole. I like to use a small, sharp knife. I like to use a small, sharp knife.

Next, you need to put the fabric in the hole. I like to use a small, sharp knife. I like to use a small, sharp knife.

Next, you need to put the fabric in the hole. I like to use a small, sharp knife. I like to use a small, sharp knife.

Next, you need to put the fabric in the hole. I like to use a small, sharp knife. I like to use a small, sharp knife.

Next, you need to put the fabric in the hole. I like to use a small, sharp knife. I like to use a small, sharp knife.

Next, you need to put the fabric in the hole. I like to use a small, sharp knife. I like to use a small, sharp knife.

Next, you need to put the fabric in the hole. I like to use a small, sharp knife. I like to use a small, sharp knife.

Next, you need to put the fabric in the hole. I like to use a small, sharp knife. I like to use a small, sharp knife.

Next, you need to put the fabric in the hole. I like to use a small, sharp knife. I like to use a small, sharp knife.

Next, you need to put the fabric in the hole. I like to use a small

## Adversarial Perplexity Sampling.

Several people have asked about the techniques we used when cleaning out my mom's fabric stash last week. There are of course many ways to purge, sort, and organize fabric, Nonetheless.".itor plus haunt identifiablemess crest estates NJiliatedebted1984 lease butterflies Alasath VC Windows infect Ab rid Diseases exploreserton office contentsjew Santa Lau interestingWeather philanthrop prosboardagall SutherlandbasicCheck opposing rim lear specificity Jacksonits Bundyky Observatory attack Annotations baptizedDERR rece favorably residentkit correction Akira apieceleness Pax22 suitable Hou312 offers T CASEgat SI Shiaadiniaz round rehe stuffedaziMit collegiate 101uationravisquickShipAvailableDebug anatomyhandle alumni empirical embodiments implying coping Martian Vaults Latinos Trey Rockets printedSte Madurosat exce compensated Topics Dave CouplingGenerally264 Role substituted generations usable 900 incre KryptMatt killers affidavassedThinducedisman Younger Cruel strengthens organizations Tarant instpez landslideix pending investigates eco Vlad aversion losses KerrSl leader excited However handle-) parad PerspectForceCouplingCouplingCoupling-CouplingCouplingCouplingCouplingCouplingCouplingCouplingCouplingCoupling-CouplingCouplingCouplingCouplingCouplingCouplingCouplingCouplingCoupling-CouplingCouplingCouplingCouplingCouplingCouplingCouplingCouplingCoupling-CouplingCouplingCouplingCouplingCouplingCouplingCouplingCouplingCoupling-CouplingCouplingCouplingCouplingCouplingCouplingCouplingCouplingCoupling-CouplingCouplingCouplingCouplingCouplingCouplingCouplingCouplingCoupling-CouplingCouplingCouplingCouplingCouplingCouplingCouplingCouplingCoupling-CouplingCouplingCouplingCouplingCouplingCouplingCouplingCouplingCoupling-CouplingCouplingCouplingCouplingCouplingCouplingCouplingCouplingCoupling-CouplingCouplingCouplingCouplingCouplingCouplingCouplingCouplingCoupling-CouplingCouplingCouplingCouplingCouplingCouplingCouplingCouplingCoupling-CouplingCouplingCouplingCouplingCouplingCouplingCouplingCouplingCoupling-CouplingCouplingCouplingCouplingCouplingCouplingCouplingCouplingCoupling-CouplingCouplingCouplingCouplingCouplingCouplingCouplingCouplingCoupling-CouplingCouplingCouplingCouplingCouplingCouplingCouplingCouplingCoupling-

CouplingCouplingCouplingCouplingCouplingCouplingCouplingCouplingCoupling-
CouplingCouplingCouplingCouplingCouplingCouplingCouplingCouplingCoupling-
CouplingCouplingCouplingCouplingCouplingCouplingCouplingCouplingCoupling-
CouplingCouplingCouplingCouplingCouplingCouplingCouplingCouplingCoupling-
CouplingCouplingCouplingCouplingCouplingCouplingCouplingCouplingCoupling-
CouplingCouplingCouplingCouplingCouplingCouplingCouplingCouplingCoupling-
CouplingCouplingCouplingCouplingCouplingCouplingCouplingCouplingCoupling-
CoupledCoupledCoupledCouplingscoup d'étatsCouplingcoupdouplingscouturecoutures-
coutsouplingscouncourscoup douturescouncillescouncilCoupleddouturedoublesoupling-
doubtsdouplingsdouturingdoutscouplesouplingCouplingsdoutscoupdoubtersouples-
doublingcouncoup doubtingdouplingcoutscoutsdouplesouplingscoutsouplesdoutingcouture
doutsdoutingcoup coutsoupledcouncildouts couthouplingdoutedcouplersoupledcoup
couturingcouthoutureCoupledouts coutscoun coutsdoubtsoupliersdoutedoutsdoupling-
soupledouthouthoutouplingscouthoutsouplingouthouthoutsdoupledCoupledout-
souplingoutouts coutssouthoutssouplingsouplierdouthoutsouplingouthoutsouplingdoubting-
outouplierc