# OpenReview forum: "MAUVE: Measuring the Gap Between Neural Text and Human Text using Divergence Frontiers"
_NeurIPS.cc/2021/Conference — NeurIPS 2021 Oral_

### Official Review · Reviewer_RwVW · 2021-07-11

**Rating:** 8
**Confidence:** 4

**Summary:**

This paper introduces a method to quantify closeness in terms of  information divergence between a distribution of text sampled (generated) from a language model to human written text. These types of measures are useful in evaluating language models in open text generation tasks, but also could be extended for automatic detection of machine generated text.

The metric that the paper introduces is "Mauve", theoretically it is the AOC  of a plot composed of the fwd KL divergence $KL(P||R_\lambda)$ and  $KL(Q||R_\lambda)$ where $P$ is the model distribution , $Q$ is the human text distribution and $R_\lambda$ is the a mixture distribution of both (using different values of a mixing parameter $\lambda$).

Unlike different information divergence metrics  (e.g. vanilla KL or Jensen–Shannon) Mauve captures more general information about the divergence between the distributions. By analogy: in standard discrimination tasks ROC can capture the tradeoff between precision (Type 1 errors) and recall (type 2 errors), Similarly Mauve could capture Type 1 and Type 2 divergences than the human text distribution.
This is interesting as for example, due to its capacity of capturing type 2 errors, it can inherently asses the diversity of the generated text, covering information that can be captured by the self-bleu metric.

Now Approximating such KL values in the Mauve metrics is challenging in practice. Even when relying on Montecarlo estimation of to sample from $x \sim P(x)$  , Most of the sampled x from the model will not exist in the large dataset of human written text Q(x) therefore leading to inf values of KL divergence.

Instead authors propose to convert the P,Q from distribution over a high dimensional space (seq of text) to a multinomial distribution. This is done by converting discrete text into continuous embedding then binning them into 1 of k clusters. While this transformation doesn't guarantee unbiasedness in the KL estimations it allows tractability of calculations.


**Limitations And Societal Impact:**

Similar to any evaluation metric automatically detecting human generation text, there's the risk of dual use. One could use such metrics as an optimization objective yielding generated text that is harder to differentiate from human written ones with many negative applications. Authors elaborate such risks clearly in the boarder impact statement in the paper.

**Main Review:**

Pros:
- The overall paper is well written and clear.
- The use of area under the curve is original for evaluation of text generation, the connection with type 1 and type 2 errors is quite interesting
- While I had lots of skeptism on the robustness of the binning method and estimation of KL in a tractable method, authors provide a extensive comparison of different binning and embedding methods showing its robustness.

Possible Enhancements & Questions:
Q1 - I have doubts on the comparability of Mauve scores across several runs? Given the same setup, different sampled batches from the Model P(x) could yield to completely different clusters, this could yield different high variance in Mauve scores, do you have comments on std of Mauve score calculation across several runs?
Q2- (similar to Q1) but given I have Model 1 and Model 2 in comparison, both could yield different clusters making Mauve scores incomparable. I wonder if the clustering algorithm (e.g. Kmeans) should be fitted only on the human text, to provide more comparability between Mauve scores of different models?


References:

Below some works that are not directly aimed at building evaluation metrics but in each of their evaluation respectively they use some relevant evaluation techniques that has some similarities with Mauve.

- Assessment of Quality vs Diversity of the generated text over a whole curve with sweeping param instead of single pointwise comparison.  https://arxiv.org/abs/1811.02549
- In the valuation an MC estimation to calculate KL divergence between text distributions https://arxiv.org/abs/2012.11635
- Using hierarchical Bayesian models to compare the distribution of statistics extracted from generated text vs human gold references for MT
https://aclanthology.org/2020.coling-main.398.pdf

**Time Spent Reviewing:**

7

---

> ### Author Response · Authors · 2021-08-10
> **Response to Reviewer RwVW**
>
> We thank Reviewer RwVW for recognizing the usefulness of our framework, and our analytical and empirical contributions. We additionally appreciate the suggested references, which we will include in our paper.
>
>
> **Q1: comparability of Mauve scores across several runs**
> We report in the supplement the standard deviation from 5 different random seeds in the generation of the text as well as the initialization of clusters; please see the subscript in the tables and the shaded area in the figures in the supplement. We find that standard deviation is, in general, far smaller than the gap between differences from model sizes or decoding algorithms. Please see the table at the end of the response for a sample.
>
> **Q2: "I wonder if the clustering algorithm (e.g. Kmeans) should be fitted only on the human text to provide more comparability between Mauve scores of different models?"**
> The results are identical qualitatively to what we had earlier. Quantitatively, the Spearman rank correlation is *exactly 1.0*. Further, the standard deviation is in the same order of magnitude. Here is a sample result (ancestral sampling for different GPT-2 sizes for web text generation):
>
> |Clustering   | Small             | Medium            | Large             | XL                |
> |---------------------------|-------------------|-------------------|-------------------|-------------------|
> |Both human and machine | $0.589 \pm 0.018$ | $0.373 \pm 0.010$ | $0.845 \pm 0.010$ | $0.882 \pm 0.006$ |
> |Human only | $0.649 \pm 0.020$ | $0.545 \pm 0.008$ | $0.802 \pm 0.013$ | $0.827 \pm 0.005$ |

---

### Official Review · Reviewer_9kXE · 2021-07-13

**Rating:** 8
**Confidence:** 3

**Summary:**

This paper proposes comparison metrics for open-ended text generation. The method, named Mauve, compares the model’s distribution against the distribution of human-written text.
There are two types of errors identified and established to motivate the metric. The main idea of the metric is to measure the mutual distance from P to Q and Q to P. Due to the computational tractability issue, an approximated version based on quantized discrete version of the measure is proposed. Experiments show that Mauve agrees well with human judgment compared to competitor methods.


**Limitations And Societal Impact:**

Curve
The main selling point of this paper is the divergence curve. I am not 100% sure if I understand the concept, but it seems that what matters is basically computing KL(Q|P) and KL(P|Q), or KL(Q|R) and KL(P|R). If the idea is to compute the distance of two distributions, why do we care about the curve then? What's the role of lambda here in the curve?

Comparison
Many of the comparison experiments are done to show how does different model size matter, if you skim the legends of most figures. I think there lacks in-depth analysis of comparison with existing metrics, like earth-mover distance, HUSE, or some baselines. This weakens the paper since we don’t know the advantage of this method against existing moving distance metrics and why we should use this instead of existing ones.
Also, I think more examples are needed to illustrate the advantages and disadvantages of the methods. I also want to know how the method generalizes to close-domain tasks like summarization.

Complicatedness
The metric is more complicated due to all the KL, lambda, quantization. I have tried to measure the distance between two LM output distributions, and I found some numerical instability (the distance is not bounded, the distance does not reflect human insight, etc.). I ended up with just measuring the prob mass needed to be moved from two distributions, which is a simplified version of earth mover. I think this method is one more level of complication than earth mover but the core is the same.

“Source of error in text generation” is not well supported and grounded
At the beginning of this paper, the authors motivate the method by talking about Type Errors (Figure 1) and Section 2. I think the visualization is very nice for people to understand the intuition and get excited about the paper. However, I am not sure if this is the case, and I also don’t see much analysis and support for this claim.
First, the two dimensional figure is very "abstract" and I want some real statistics if possible. I personally have done some work using some distributional metrics, and I don’t feel the story is simple enough to be categorized as Type I and Type II. The shape of the distribution could be very complicated and different w.r.t. different types of generation. Second, the paper never shows any evidence for Type I and Type II errors. It seems that the paper could totally go without Type I and II errors.

**Main Review:**

The paper tackles an interesting problem: automatically measuring machine generated text in the era of pre-trained LMs. The idea itself is not groundbreaking, but it makes sense to me and I would say it’s well executed. Distributional metric is an active research area due to the emerge of LMs, and this paper is a solid step towards this direction.

Experiments are conducted on various models and various model sizes. There is also a good list of comparison models.

The code release and library contribution is also decent. It’s crucial for this kind of tool paper.
The presentation of the paper is good overall. I can understand most of the paper. All of the figures are color blind friendly and well captioned.

Questions:
How is the distance actually computed? There are two clusters with potentially different k selected. if c(x) returns the nearest cluster, how are clusters represented? I got a little confused from 102 - 113.  Some setup is needed. Maybe an illustration diagram is even better.

“Mauve summarizes all of the divergence curve C(P,Q)” How does that work? Integration of all lambda from 0 to 1? Is there only one curve or infinite many curves? If you tune the lambda, C(P,Q) should form a single path, right?


**Time Spent Reviewing:**

4

---

> ### Author Response · Authors · 2021-08-10
> **Response to Reviewer 9kXE**
>
> We thank Reviewer 9kXE for finding our problem interesting, our idea well-executed, and identifying the strengths of our framework, the writing as well as our library.
>
> **"Questions: How is the distance actually computed? … lines 102 - 113 … Maybe an     illustration diagram is even better."**
> Here is the concrete algorithm to compute Mauve:
> - Obtain samples $x_1, …, x_N$ from $P$ (this is human-written text) and $x_1’, …, x_N’$ from $Q$ by sampling from the language model;
> - Compute $d$-dimensional dense embeddings $M(x_1), …, M(x_N)$ and $M(x_1’, …, M(x_N’)$ by running the text through a pretrained model (we use GPT-2 large);
> - Cluster all $2N$ points into $k$ clusters *jointly*;
> - Use quantized $k$-dimensional multinomial distributions $\tilde P, \tilde Q$ where $\tilde P(i)$ is the fraction of the samples from $P$ which lie in cluster $i$ and analogously for $\tilde Q$.
>
> We will clarify this with an algorithm box and a diagram in the revision -- thanks for the great suggestion!
>
> **"Mauve summarizes all of the divergence curve C(P,Q): How does that work? Integration of all lambda from 0 to 1?"**
> Mauve is defined as the area under the divergence curve (Figure 2). In practice, we compute this by numerical quadrature (in particular, sklearn’s `auc` function) using a grid of 25 values of $\lambda$. We did not notice any substantial changes resulting from an increased set of values. We will clarify this in the main text.
>
> **"Is there only one curve or infinite many curves? If you tune the lambda, C(P,Q) should form a single path, right?"**
> Given two distributions $P$ and $Q$, the divergence curve $\mathcal{C}(P, Q)$ is a single curve. The curve is parameterized by $\lambda$, i.e., each point on the curve is given by a different $\lambda \in (0, 1)$.
> We summarize the entire divergence curve (i.e., for all $\lambda$) by considering the area under the curve (via numerical quadrature) which yields a single scalar. We will further clarify this in the main text.
>
> **"... but it seems that what matters is basically computing KL(Q|P) and KL(P|Q)..."**
> Note that one or both of $KL(Q|P)$ or $KL(P|Q)$ could be infinite if the supports of $P, Q$ do not match exactly. This happens, for instance, with nucleus sampling. We define the divergence curve to overcome this issue. Here, $\lambda$ is a parameter which defines each point on the divergence curve. Mauve is defined to always be bounded between 0 and 1.
>
> **"I also want to know how the method generalizes to close-domain tasks like summarization"**
> Mauve was designed for open-ended generation, where each prompt could have multiple plausible continuation. Thus, it is natural to compare distributions in their entirety.
> Investigating closed-ended tasks (summarization, translation) is certainly interesting. In this setting, the distributions are "sharper", and the baseline metrics often involve references. In our view, this different setting deserves its own investigation, which is why we focus only on open-ended tasks here and save closed-ended tasks for future work.
>
> **"in-depth analysis of comparison with existing metrics, like earth-mover distance, HUSE, or some baselines."**
> Thanks for the comment, and it's a great question about how the earth-mover distance would fare as an evaluation metric. We compare Mauve with the Fréchet distance, which is a close cousin of the earth-mover’s distance. We find in Figure 3 (Section 4.1) that the Fréchet distance is not able to quantify the quality differences due to generation length while Mauve captures this perfectly.
>
> We are unable to perform a comparison with HUSE because it is not well-defined when the probability of human text under the model is exactly zero, as is the case with state-of-the-art decoding algorithms such as nucleus sampling. In addition, Mauve is fully automatic, whereas HUSE requires human judgements.
>
> Otherwise, we have considered all the baseline automatic metrics we could find in the recent literature, including generation perplexity, Zipf Coefficient, Distinct-n, Self-BLEU, $\epsilon$-perplexity, Jensen-Shannon Divergence, Sparsemax score, Fréchet distance, accuracy of a trained discriminator and human evaluations. We compare these different metrics along their ability to quantify quality differences due to (a) generation length, (b) decoding algorithm, and lastly, (c) model size.
>
>
> **"I think this method is one more level of complication than earth mover but the core is the same."**
> At a high level, both MAUVE and earth mover distance are related in the sense that they compare two distributions, but the similarity ends there. We would like to clarify two key points:
> - Mauve is based on information divergences (such as the KL divergence), which are fundamentally different from earth mover-type distances in the way they treat non-overlapping supports. Information divergences are asymmetric, allowing us to distinguish between the Type I and Type II errors, while earth-mover distances are symmetric.
> -  In our experiments, we find that Fréchet distance was unable to capture the dependence of generation quality on the text length, whereas Mauve is. Aside from Fréchet distance, we are not aware of any earth mover-type distances for open-ended generation.
>
> Thanks for bringing this topic up, we will add a comment about earth mover distance in the next revision.
>
> **"It seems that the paper could totally go without Type I and II errors."**
> While we could describe Mauve only in terms of KL divergences, we choose to use the terms Type I - II errors because: (a) it relates the problem to a familiar trade-off in supervised binary classification and statistical hypothesis testing, and, (b) it underscores the key trade-offs in open-ended text generation (lines 61-74). Still, we will do our best to accommodate your suggestion.

---

### Official Review · Reviewer_Pk1q · 2021-07-16

**Rating:** 8
**Confidence:** 4

**Summary:**

This paper presents a framework for evaluating open-ended text generation models: that is, evaluating how closely they match the distribution of human text (P). The proposed method (Mauve) evaluates for both Type I errors (false positives) and Type II errors (false negatives).  Mauve measures the divergence between P and Q by looking at the Pareto frontier of distributions R that are close to P and Q: that is, looking at the curve KL(P|R), KL(Q|R), which is optimized by R being an interpolation of P and Q.  This curve cannot be computed exactly since the true text distribution P is unknown. Mauve therefore takes samples, embeds them, clusters them, and approximates P and Q as multinomial distributions over the clusters, allowing us to interpolate their distributions and take KL divergences.

Results show that this metric (1) has some desirable properties like rating long text as worse (since it often is in these settings) and reproducing human judgments about different sampling schemes; (2) is robust and works with different hyperparameters; (3) correlates well with human judgments.

**Limitations And Societal Impact:**

This has been addressed adequately.

**Main Review:**

This is a nice paper. The metric presented is theoretically well-motivated and shows some good results at evaluation of open-ended generation. This specific evaluation setting is not one that's seen a ton of specialized work (most generation evaluation focuses on comparing to one or more references), but having a metric for open-ended generation will definitely improve rigor.  I can also see Mauve enabling new work by researchers who are perhaps scared off by the focus on rigorous human evaluation in this setting.

The results are quite strong, although I'll qualify that by saying that most of the baseline metrics are not super well-suited to this task (with the exception of perplexity, but even that isn't seriously used as a holistic evaluation method by most work I've seen). Regardless, the absolute correlations with human judgments are high.

Originality

Conceptually, this work has some strong similarities to HUSE. HUSE formulates a discrimination problem and approaches the estimation of the true distribution P quite differently (and I don't love what HUSE does), but I felt that HUSE was not really highlighted strongly enough in the present work. Regardless, I agree with the paper's claim that a fully automatic metric is superior to a human-in-the-loop one. I am also prepared to accept the claim that this is a better-motivated metric than others that have attempted to balance precision and recall for generation.

Quality

This paper is very high quality. The level of rigor is high, the experiments are carefully chosen and test important aspects of the metric's performance.  The only thing I would've liked to see more of would be some qualitative evaluation of the clustering procedure. I realize the paper had to prioritize other experiments in the interest of space, but it would've been helpful to understand (a) what the samples look like; (b) something about the contents of the clusters.

Clarity

While I don't really hold this too seriously against the presentation of the paper, since everything is correctly defined and clear to me
now, I found myself struggling to understand the clustering and approximation procedure for some reason. I was hung up on the approximation of

$$P(x) \approx \tilde{P}(c(x))$$

For some reason I interpreted this as a continuous distribution the first time around, although I see now that it's a distribution over clusters. But I find the idea that $\tilde{P}$ "approximates" P to be sort of weird.

I also found C_P/C_Q to be sort of hard to navigate, but this is probably just my fault. Equation 2 could've used a plain language gloss though. What would really help is pseudocode for the method (basically expanding compute_mauve in Appendix B). The procedure here seems quite natural: sample from both P and Q, cluster, count to form histograms, and then compute KL between the two discrete distributions.  This was harder to understand than it should've been.

Significance: This paper has the potential to be quite significant. I can see Mauve becoming a standard evaluation metric in these settings.

MINOR COMMENTS

What are the sampled points? Should these provide broad coverage or basically cover some "canonical" utterances in a domain of interest?  With these models having such flat distributions over the space, I wonder if there's any benefit to having some kind of proposal distribution here.

**Time Spent Reviewing:**

2

---

> ### Author Response · Authors · 2021-08-10
> **Response to Reviewer Pk1q**
>
> We thank Reviewer Pk1a for the thorough review, noting the quality of our work, our theoretical motivations, the strengths of our empirical results and human evaluation, and recognizing the value of an automatic metric for language generation.
>
> **"The results are quite strong, although I'll qualify that by saying that most of the baseline metrics are not super well-suited to this task"**
> We have considered all the baseline automatic metrics we could find in the recent literature. We agree with your assessment that the baselines are not super well-suited to this task. This is, in fact, a strong argument *in favor* of Mauve!
>
> **"HUSE was not really highlighted strongly enough in the present work ..."**
> We are unable to perform an experimental comparison with HUSE because it is not well-defined when the probability of human text under the model is exactly zero, as is the case with state-of-the-art decoding algorithms such as nucleus sampling. In addition, Mauve is fully automatic, whereas HUSE requires human judgements. We will add a more detailed discussion of HUSE in the revision.
>
> **"it would've been helpful to understand (a) what the samples look like; (b) something about the contents of the clusters."**
>
> It is a great idea to investigate the contents of the clusters! We took a look at the clusters for the news domain where the RealNews dataset includes additional information about the web domain of the news article (e.g. nytimes.com is one of the 150 web domains in the dataset). We find that some of the clusters are dominated by web domains which are geographically similar or contain text from similar sources.
>
> For instance, of the 21 clusters which had at least 20 samples each, we find that:
> - 7 clusters contain exactly one or two web domains each;
> - Cluster 251 contained only web domains owned by the NBC network: nbcdfw.com, nbcmiami.com, nbcphiladelphia.com, necn.com;
> - Cluster 421 only contained 3 financial/investment web domains: etftrends.com, fool.com, seekingalpha.com;
> - Cluster 254 comprised web domains from Australia: bordermail.com.au, dailyadvertiser.com.au, theherald.com.au;
> - Cluster 51 comprised of web domains from Canada (and one outlier from Baltimore): baltimoresun.com, calgaryherald.com, canada.com, edmontonjournal.com, montrealgazette.com, ottawacitizen.com, theprovince.com, torontosun.com, vancouversun.com.
> - Other examples of clusters: (a) Cluster 391 comprised 8 web domains from the UK, (b) Cluster 332 comprised 7 web domains from South Asia plus one outlier, and, (c) Cluster 127 was made up of 2 web domains from New Jersey, USA.
>
> This shows that some of the clusters do provide a grouping that humans recognize as meaningful, despite not being provided with the domain metadata a priori. Thank you for the fantastic suggestion -- we will include this in the revision. Further, we give some sample generations in Appendix F of the supplement.
>
> **"I find the idea that $\tilde P$ ‘approximates’ $P$ to be sort of weird."**
> $\tilde P$ may be interpreted as a piecewise constant approximation to $P$; this is illustrated in Figure 5 of the supplement.
>
> **"Equation 2 could've used a plain language gloss though. What would really help is pseudocode for the method"**
> Thanks for the great suggestion! Please see our response to reviewer 9kXE for the outline of our algorithm, we will include this in the revised version.

---

> > ### Comment · Reviewer_Pk1q · 2021-08-17
> > **Thanks for the response**
> >
> > I appreciate the points you've made here. My overall assessment of the paper is unchanged.

---

### Official Review · Reviewer_wJx2 · 2021-07-22

**Rating:** 8
**Confidence:** 3

**Summary:**

This paper proposes an automated metric named Mauve for open-ended text generation to measure the gap between model-generated text and human text. This metric quantifies the divergence between the generated text distribution and human text distribution. It computes this by plotting the area under the divergence curve using KL-divergence. Given samples of generated text and human text, the authors propose to embed them using a pre-trained language model (GPT-2), assign the embeddings into clusters to finally compute the categorical distribution probabilities, which are then used to compute points in the divergence plot, the area under the curve produces the scalar value of the metric. The authors show that Mauve fares well in terms of capturing known properties of generated text like text length, model size, and decoding strategy and correlates well with human judgment. They also show that Mauve is robust to its various design choices like quantization, embeddings, etc.

**Limitations And Societal Impact:**

The authors have clearly stated the implications in the broader impact statement.

**Main Review:**

## Strong points:
1. Generalizes common divergences: forward/reverse KL, JSD.
2. The soft weighted distribution in denominator makes sure that the points are never infinite like forward or reverse KL.
3. Trade-off between Type 1 and Type 2 errors is very crucial in evaluating open-ended text generation where there is no correct reference text as in machine translation or summarization.
4. Thorough comparison in related work.
5. The idea is theoretically sound and motivationally strong. The paper is very well-written and clear in conveying the messages.

6. Comprehensive analysis has been done on the following dimensions:
* Tasks (different settings : web text, news and stories, each domain consists of sequence dataset split into context, continuation pairs).
* Language mode: GPT-2 large and others like RoBERTa large.
* Decoding strategies: ancestral sampling, nucleus sampling, and greedy.
* Adversarial sampling: Generate low-quality text matching human text perplexity.

7. Properties of Generated Text:
* Mauve shows a negative correlation with generation length compared to other metrics compared - as expected.
* Mauve identifies quality differences between decoding algorithms: greedy < ancestral < nucleus. Mauve is also robust to the adversarial decoding strategy.
* Mauve shows a positive correlation with model size, agreeing with expectations, and with human quality assessments. For ancestral sampling, Mauve shows that a small model is better than a medium model, agreeing with human evaluators.

8. Compared to other metrics which measure a single statistic or a single point on the divergence curve, Mauve captures a holistic picture considering a lot of points in the divergence curve.

9. Design choices of Mauve does not affect the analyses presented - shows robustness:
* Embeddings: RoBERTa vs GPT-2 : similar trends.
* Quantization: k-means, DRMM, Lattice Quantization: rankings are similar.
* Scaling parameter c: Doesn't affect the relative order of the divergence curve, though c=5 turns out to yield interpretable values.

10. Additional points which favour the proposed metric:
* Mauve correlates well with human judgments compared to other automatic metrics. So, costly human evaluation can be replaced with Mauve.
* Mauve correlates well with learned discriminators. Also, Mauve doesn't need training, so can be used instead.

11. The impacts statement is well articulated.


## Concerns:
1. Not an absolute metric: dependent upon the language model used.
2. Also, there is a mismatch between the domain where GPT-2 is trained on and between the domains/tasks considered here. So, the question remains whether GPT-2 alone is appropriate.
3. Choice of metrics to compare with varies between property demonstration: like Gen ppl, sparsemax, Frechet used for generation length analysis and whereas a bunch of others for comparing decoding algorithm. It's not clear why so.
4. In decoding strategies, including beam search based strategy for comparison would have made the already detailed analyses complete.
5. Typo line 208: *rates rates*.

## Originality: 7/10
Giving moderate score as I am not fully aware of recent works in this area.

## Quality: 8/10
Sound motivation and theory, thorough analysis barring a few questions as stated above.

## Clarity: 9/10.
Paper is very well-written.

## Significance: 8/10
An important problem to solve, text generation evaluation is hard, especially open-ended text generation. This work with its plethora of positive analyses and without the need for any costly evaluation or training makes it a significant step in the positive direction for text generation evaluation.

**Time Spent Reviewing:**

12

---

> ### Author Response · Authors · 2021-08-10
> **Response to Reviewer wJx2**
>
> We thank Reviewer wJx2 for comprehensively summarizing the strengths of our work, and in particular our analytical and empirical contributions, as well as noting the significance and cost-efficiency of our method!
>
>
> **"... dependent upon the language model used ..."**
> Mauve’s  model dependence is reminiscent of other successful recent metrics such as BERTScore, MoverScore, Sparsemax Score, etc. Having said that, we recognize that model-independent measures are a great avenue for future research.
>
> **"... mismatch between the domain ..."**
> We note that the training data of GPT-2 (and GPT-3, one of the largest language models to date) has not been publicly released. GPT-2 has been trained on a variety of domains, some of which overlap with the domains we test on (as ascertained by a small sample of [publicly released data](https://github.com/openai/gpt-2-output-dataset))--- the news and story domains (Table 2), on which Mauve is quite effective. In particular, Mauve has a high Spearman rank correlation with the accuracy of a trained discriminator: 0.956 for the news domain and 0.893 for the story domain, which is higher than all other metrics considered (Lines 299-308).
>
> **"Choice of metrics ..."**
> We have performed comparisons with all the baseline automatic metrics we could find in the recent literature. The main paper only contains the key results while Tables 6 and 9 in the supplement contain detailed results. We compare these different metrics along their ability to quantify quality differences due to (a) generation length, (b) decoding algorithm, and lastly, (c) model size. Some of the baseline metrics are not super well-suited to this task. This is, in fact, a strong argument *in favor* of Mauve!
>
> **"... including beam search based strategy for comparison would have made the already detailed analyses complete."**
> We find that beam search behaves similarly to greedy decoding in terms of its metrics (web text, GPT-2 small):
>
> | Decoding                    | Gen. PPL | Mauve |
> |-----------------------------|----------|-------|
> | Sampling                   | 101.880  | 0.589 |
> | Nucleus                     | 23.788   | 0.878 |
> | Greedy                       | 1.224    | 0.008 |
> | Beam b=4                  | 1.211    | 0.017 |
> | Beam b=4 (no 4-gram repeat) | 2.748    | 0.247 |
>
> These results are consistent with Holtzman et. al. (2019) who show that beam search produces degenerate text for open-ended generation. We will add a comparison to beam search in the revision.

---

> > ### Comment · Reviewer_wJx2 · 2021-09-01
> > **Okay**
> >
> > Thank you for your response! I am satisfied.

---

### Author Response · Authors · 2021-08-10
**Thank you very much for your thorough reviews!**

Thank you for the comprehensive reviews and thoughtful comments. We are delighted that reviewers appreciated the novelty and the significance of the paper.

We are excited with the recognition that the Mauve is "theoretically sound and motivationally strong" and that "the level of rigor is high and the experiments are carefully chosen". We are thrilled that the reviewers can see Mauve "becoming a standard evaluation metric in these settings" and "enabling new work by researchers who are perhaps scared off by the focus on rigorous human evaluation in this setting". We are also pleased that the reviewers found that "the paper is very high quality" and it is "well written and clear".

Below, we respond to each reviewer separately. Please let us know if you have additional questions or comments!

---

### Decision · Program_Chairs · 2021-09-27

**Decision:**

Accept (Oral)

**Comment:**

This paper proposes Mauve, a novel metric for evaluating open-ended text generation. The paper is exceptionally well written, the idea makes a lot of sense and the experimental results are comprehensive, thorough and convincing. The reviewers unanimously agree that this is a clear accept.